# Influenza infection rewires energy metabolism and induces browning features in adipose cells and tissues

Asma Ayari[1], Manuel Rosa-Calatrava[2,5], Steve Lancel[3,5], Johanna Barthelemy[1], Andrés Pizzorno [2], Alicia Mayeuf-Louchart[3], Morgane Baron[1], David Hot [1], Lucie Deruyter[1], Daphnée Soulard[1], Thomas Julien[2], Christelle Faveeuw [1], Olivier Molendi-Coste[3], David Dombrowicz[3], Laura Sedano[4], Valentin Sencio [1], Ronan Le Goffic [4], François Trottein[1] & Isabelle Wolowczuk [1]✉

Like all obligate intracellular pathogens, influenza A virus (IAV) reprograms host cell's glucose and lipid metabolism to promote its own replication. However, the impact of influenza infection on white adipose tissue (WAT), a key tissue in the control of systemic energy homeostasis, has not been yet characterized. Here, we show that influenza infection induces alterations in whole-body glucose metabolism that persist long after the virus has been cleared. We report depot-specific changes in the WAT of IAV-infected mice, notably characterized by the appearance of thermogenic brown-like adipocytes within the subcutaneous fat depot. Importantly, viral RNA- and viral antigen-harboring cells are detected in the WAT of infected mice. Using in vitro approaches, we find that IAV infection enhances the expression of brown-adipogenesis-related genes in preadipocytes. Overall, our findings shed light on the role that the white adipose tissue, which lies at the crossroads of nutrition, metabolism and immunity, may play in influenza infection.

[1] University of Lille, CNRS, INSERM, CHU Lille, Institut Pasteur de Lille, U1019 - UMR 8204 - CIIL - Center for Infection and Immunity of Lille (CIIL), Lille, France. [2] Laboratoire Virologie et Pathologie Humaine - VirPath Team, Centre International de Recherche en Infectiologie (CIRI), INSERM U1111, CNRS UMR5308, Ecole Normale Supérieure de Lyon, Université Claude Bernard Lyon 1, Université de Lyon, Lyon, France. [3] University of Lille, INSERM, CHU Lille, Institut Pasteur de Lille, U1011 - EGID, Lille, France. [4] Laboratoire Virologie et Immunologie Moléculaires (VIM), INRA, Université Paris-Saclay, Jouy-en-Josas, France. [5] These authors contributed equally: Manuel Rosa-Calatrava, Steve Lancel. ✉email: isabelle.wolowczuk@pasteur-lille.fr

Influenza A viruses (IAVs) are important human respiratory pathogens that lead to substantial morbidity and severe disability during annual epidemics[1]. Older adults and pediatric populations are particularly at risk for influenza complications. Influenza viruses can also cause recurrent, sporadic and potentially devastating pandemic outbreaks, which create enormous health and economic burdens worldwide. Despite great advances in our understanding of the mechanisms of IAV infection, there is still a major need for more effective virus control modalities; achieving this goal will require a better understanding of the complex virus-host interactions.

An IAV infection starts in the respiratory tract, where viruses must attach to, invade and replicate in airway epithelial cells before spreading to neighboring non-immune cells and innate and adaptive immune cells[2]. The antiviral innate-immune response has several major roles in host defense: it limits (or at least controls) viral replication during the early phase of infection, and induces the subsequent adaptive immune response that plays a critical role in viral clearance and development of long-lasting immunity during the later stages of infection[3]. Although influenza infection predominantly results in damage to the respiratory system, various cardiovascular, neurological, and gastrointestinal complications have also been reported[4]. Even though the data are subject to debate, it has been suggested that highly pathogenic IAVs can replicate outside the respiratory system—notably in the intestine[5,6].

The link between influenza infection and metabolic disorders remains elusive. On one hand, data from animal and human studies have identified type 2 diabetes and obesity (both of which are characterized by chronic subclinical inflammation and hyperglycemia[7]) as risk factors for severe influenza outcomes[8,9]. On the other hand, a few case reports have suggested that influenza infection per se may trigger, unmask or aggravate metabolic disorders[4].

At the cellular level, influenza infection dramatically alters host cell's metabolism and thus creates an environment that is propitious for viral replication. Pioneering studies in the 1950s and 1960s observed greater glucose uptake and glycolysis in infected chick embryo cells[10]; in contrast, the inhibition of glucose metabolism significantly reduced viral replication[11]. More recently, Smallwood et al. reported elevated glucose metabolism in the lungs of IAV-infected patients and in in vitro-infected primary human bronchial epithelial cells[12]. Furthermore, influenza infection increases lipid and cholesterol biosynthesis in the host pulmonary cells[13,14]; accordingly, the pharmacological inhibition of fatty acid biosynthesis suppresses virus replication[15].

White adipose tissue (WAT) is a key metabolic organ. It has a major role in the regulation of whole-body glucose and lipid metabolism[16]. In addition to mature, lipid-filled adipocytes, WAT also contains stromal vascular fraction cells (composed of multiple cell types, including adipocyte progenitors (preadipocytes), and innate and adaptive immune cells) whose number and activation status can change as a function of physiological and pathological conditions, such as obesity[17–20]. As well as being the body's main energy reservoir (triglycerides stored in adipocytes), WAT is acknowledged to be a highly active endocrine tissue that secretes numerous metabolites and hormones[18]—some of which (leptin) have immunoregulatory functions[21]. Unlike other organs, WAT is compartmentalized into discrete depots distributed throughout the body. The subcutaneous adipose tissue (SCAT) and the visceral adipose tissue are the most frequently defined and studied depots. Importantly, these depots differ markedly in their metabolic and endocrine functions, and in their associations with the development of metabolic diseases[22].

With the exception of the report by Nishimura et al. showing that IAV (H5N1) can target WAT in vivo[23], the impact of influenza infection on the WAT of lean mice has not yet been investigated. Here, we found that (intranasal) IAV infection (seasonal H3N2 subtype) leads to profound changes in the major fat depots (namely, the inguinal SCAT and the visceral (epididymal) adipose tissue (EWAT)), some of which are common to SCAT and EWAT while others are unique to the SCAT, such as the appearance of thermogenic brown-like/beige adipocytes. Importantly, viral RNA- and viral protein-harboring immune cells and (less frequently) preadipocytes were detected in the WAT's stromal vascular fraction of infected animals. Using in vitro approaches, we showed that IAV replication is abortive in preadipocytes, and that infection yields brown-like/beige features in these cells, as evidenced by elevated expression of brown-like/beige specific genes. Transcription analyses further revealed that IAV infection was associated with unique metabolic rewiring in the SCAT of infected mice, and in in vitro-infected preadipocytes. Strikingly, infected mice display alterations in whole-body glucose metabolism that persist long after infection has been resolved.

Collectively, our findings may advance our understanding of the ramifications of influenza A infection biology, and may bring novel aspects of metabolic control of the complex virus–host interactions.

## Results

**Influenza alters the functions of adipose tissue.** C57BL/6 mice were intranasally inoculated with a sublethal dose of H3N2 IAV or with PBS (mock). At 7 days post-infection (7 dpi, corresponding to the peak inflammatory response in the lungs), IAV-infected mice displayed significant body weight loss (Fig. 1a) and reduced inguinal (SCAT) and epididymal (EWAT) fat mass (Fig. 1b). It is noteworthy that the mass of SCAT was positively correlated with the mass of EWAT in both mock-treated and IAV-infected animals (Supplementary Fig. 1); this indicates that influenza infection did not lead to major changes in body fat distribution. As an energy storage organ, adipose tissue stores neutral triglycerides through the lipogenic pathway and releases fatty acids through the lipolytic pathway[16]. Lipolysis of triglycerides requires three consecutive steps involving at least three different enzymes: adipose triglyceride lipase (ATGL), hormone-sensitive lipase (HSL), and monoglyceride lipase (MGL)[24]. Influenza infection resulted in the marked repression of *Atgl* and *Hsl* transcription in SCAT and EWAT. Strikingly, *Mgl* transcription was suppressed in EWAT but enhanced in SCAT. In both fat depots, infection was associated with decreased expression of lipogenic genes, such as those encoding glucose transporter 4 (*Glut4*), malic enzyme 2 (*Me2*), and fatty acid synthase (*Fasn*) (Fig. 1c).

Adipose tissue is composed of several cell types that include mature adipocytes, preadipocytes, and a range of innate and adaptive immune cells[17–20]. As an endocrine organ, WAT secretes numerous cytokines, metabolites, and hormones that originate from all these cell types[18]. By preserving the physiological in vivo cross-talk between cells, explant cultures of WAT allow analyzing the secretory function of the whole tissue[25]. We therefore used tissue explant cultures to investigate whether influenza infection was associated with changes in SCAT and EWAT's secretory function (Fig. 1d). When compared with controls, the adipose tissues collected from infected mice produced higher levels of the proinflammatory cytokines IL-1β and IL-6, and of the anti-inflammatory cytokine IL-10. Following IAV infection, the release of leptin—a WAT-derived hormone that regulates feeding behavior, body weight[26], and innate and

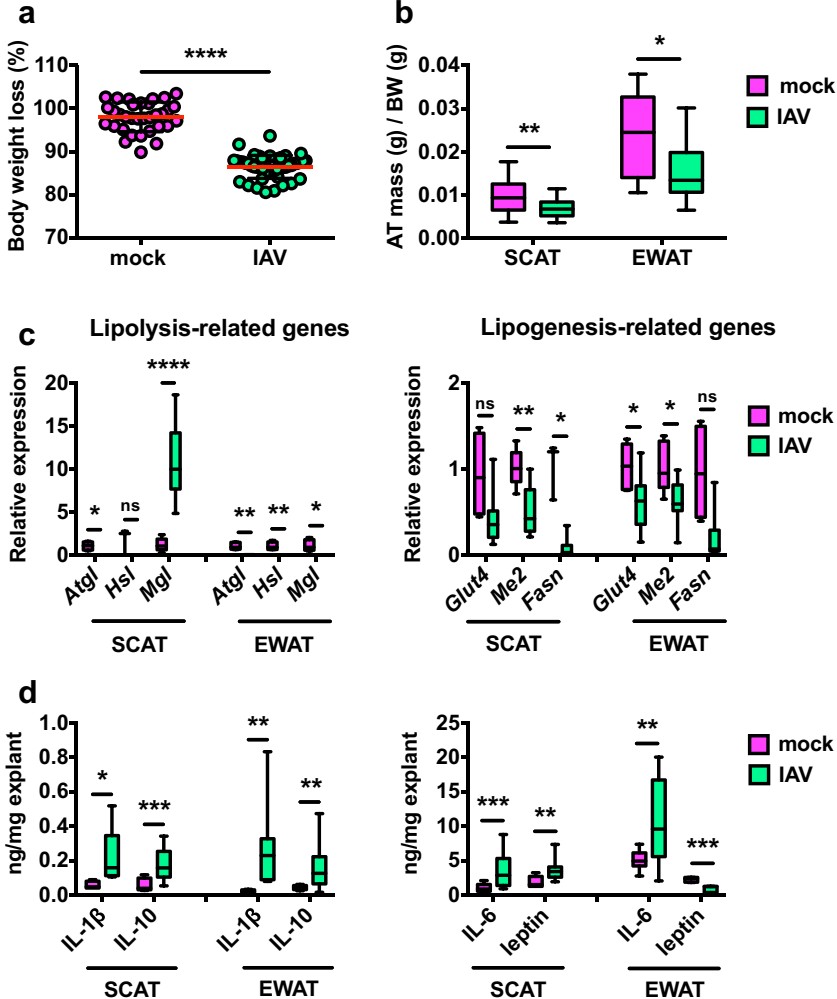

**Fig. 1 Influenza alters adipose tissues' functions. a** Body weight loss (% from initial body weight (BW)) of mock-treated and IAV-infected mice, at day 7 post-infection (7 dpi), $n = 33$ mock-treated animals and $n = 46$ IAV-infected animals. Individuals values, as well as means ± SD are shown. ****$p < 0.0001$. **b** Adipose tissue (AT) mass (g/g BW) of mock-treated and IAV-infected mice at 7 dpi, $n = 25$ mock-treated animals and $n = 28$ IAV-infected animals. *$p < 0.05$, **$p < 0.01$. **c** Real-time quantitative PCR (RT-qPCR) analysis of the expression of lipolytic genes (*Atgl*, *Hsl*, *Mgl*) and lipogenic genes (*Glut4*, *Me2*, *Fasn*) in SCAT and EWAT from mock-treated and IAV-infected mice at 7 dpi. For lipolytic genes: $n = 5$ mock-treated animals and $n = 8$ IAV-infected animals. For lipogenic genes: $n = 5$ mock-treated animals and $n = 18$ IAV-infected animals. RT-qPCR data were normalized to *Eef2* housekeeping gene expression and expressed relative to the expression obtained in the samples from mock-treated mice. *$p < 0.05$, **$p < 0.01$, ****$p < 0.0001$, ns = not significant. **d** Determination of IL-1β, IL-10, IL-6, and leptin levels (by ELISA) in the supernatants of SCAT and EWAT explants from mock-treated and IAV-infected mice at 7 dpi. For IL-1β and leptin: $n = 7$ mock-treated animals and $n = 13$ IAV-infected animals. For IL-10 and IL-6: $n = 14$ mock-treated animals and $n = 20$ IAV-infected animals. Data were normalized to the mass of the explants (mg). *$p < 0.05$, **$p < 0.01$, ***$p < 0.001$. Differences between mock-treated and IAV-infected groups (**a–d**) were considered significant when $p < 0.05$.

adaptive immune responses[21]—was enhanced for SCAT and decreased for EWAT.

Thus, IAV infection was associated with marked changes in metabolic and inflammatory features of the subcutaneous and visceral adipose tissues. Importantly, infection was associated with depot-specific modifications in WAT's lipid metabolism and leptin secretion.

**Influenza induces subcutaneous adipose tissue browning.** To determine whether SCAT and EWAT undergo qualitative and/or quantitative changes following influenza infection, we performed histologic and histomorphometric analyses at 7 dpi. The EWAT samples from mock-treated and IAV-infected mice had a similar histological appearance: white adipocytes containing large uni-locular lipid droplets (a typical WAT histology) (Supplementary Fig. 2a). In contrast, the SCAT samples from mock-treated and

IAV-infected mice showed histological differences: numerous pockets of dense, small and multilocular adipocytes—a feature of brown-like adipocytes, also known as beige or brite adipo-cytes[27]—were found interspersed within the SCAT from infected animals (Fig. 2a). As reported in literature[28], histomorphometric analyses confirmed that adipocytes are smaller in SCAT than EWAT (Supplementary Fig. 2c). In accordance with the appear-ance of small brown-like/beige adipocytes, the SCAT from IAV-infected mice showed higher numbers of small adipocytes and reduced numbers of large adipocytes than the SCAT from mock-treated mice (Fig. 2b). The differences in adipocyte size between mock-treated and IAV-infected mice were less pronounced for the EWAT (Supplementary Fig. 2b). Moreover, levels of the mitochondrial uncoupling protein 1 (UCP1, a marker of the brown adipose tissue[29]) were higher in SCAT protein lysates from infected mice than in those from mock-treated controls. In contrast, no UCP1 was detected in the lungs or EWAT lysates

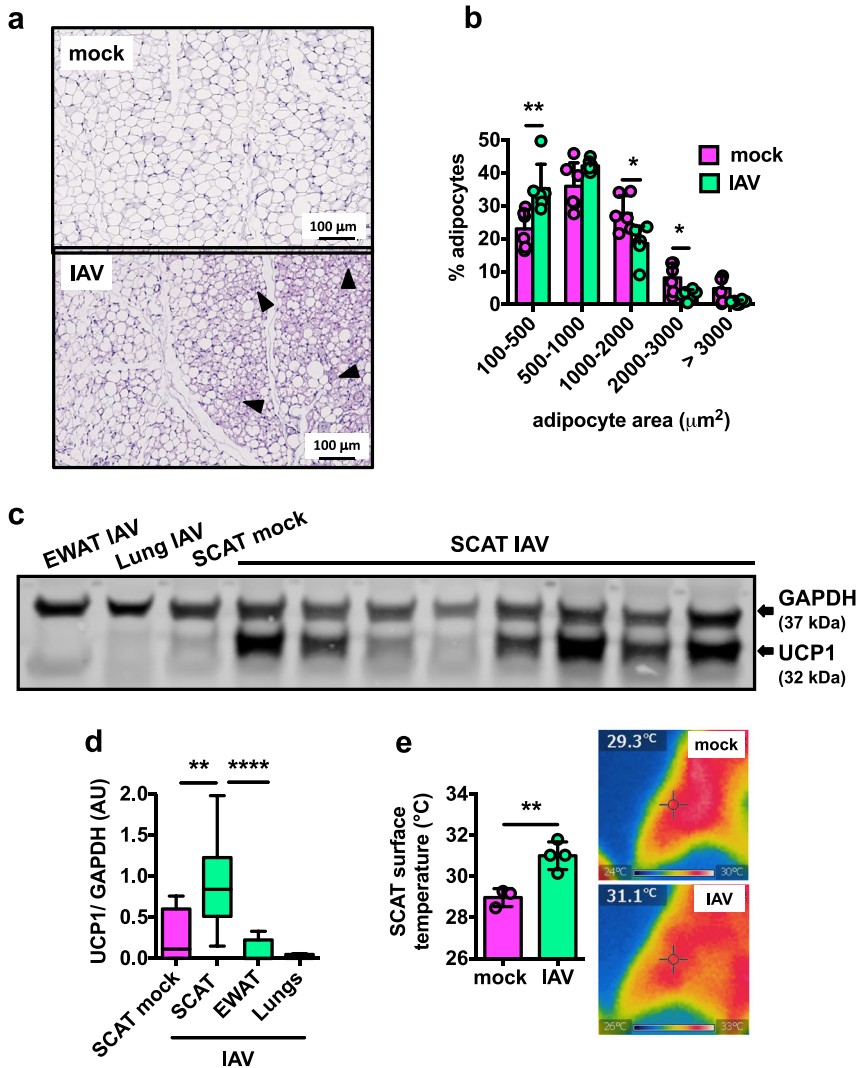

**Fig. 2 Influenza induces subcutaneous adipose tissue browning. a** Representative images of hematoxylin/eosin staining of SCAT sections from mock-treated and IAV-infected mice at 7 dpi, $n = 7$ mock-treated animals and $n = 7$ IAV-infected animals. Brown-like/beige adipocytes are arrowheaded. **b** Frequency distributions (%) of adipocyte sizes (areas, $\mu m^2$) in the SCAT from mock-treated and IAV-infected mice, 7 dpi, $n = 6$ mock-treated animals and $n = 6$ IAV-infected animals. Individuals values, as well as means ± SD are shown. *$p < 0.05$, **$p < 0.01$. **c** Western blot of UCP1 in lanes loaded with EWAT, lung, and SCAT protein lysates from mock-treated and IAV-infected mice at 7 dpi. GAPDH was used as an internal loading control. The figure is representative of four independent blots covering a total of: SCAT from $n = 5$ mock-treated animals (SCAT mock), SCAT from $n = 35$ IAV-infected animals (SCAT IAV), EWAT from $n = 5$ IAV-infected animals (EWAT IAV), and lungs from $n = 5$ IAV-infected animals (Lung IAV). Of note, the uncropped version of the blot is shown on Supplementary Fig. 2d. **d** Quantification of normalized densitometric UCP1/GAPDH ratios (arbitrary units) of all western blots. **$p < 0.01$, ****$p < 0.0001$. **e** Infrared thermography measurement (°C) of SCAT surface temperature at 7 dpi, $n = 3$ mock-treated animals and $n = 4$ IAV-infected animals. Individuals values, as well as means ± SD are shown. **$p < 0.01$. Representative infrared images of the SCAT area from mock-treated and IAV-infected mice are shown. Differences between mock-treated and IAV-infected groups (**b**, **d**, **e**), and between SCAT IAV and EWAT IAV (**d**) were considered significant when $p < 0.05$.

from IAV-infected mice (Fig. 2c, d and Supplementary Fig. 2d). In line with increased levels of thermogenic UCP1, the surface temperature generated from the SCAT was higher in IAV-infected mice than in mock-treated mice (Fig. 2e).

These results emphasized the depot-specific nature of the effects of influenza infection on fat, with the appearance of thermogenic brown-like/beige adipocytes in the subcutaneous depot being the most notable.

**Transcriptomics show fat-depot-specific response to influenza**. To gain insights into the molecular basis of the depot-specific effects of influenza infection on fat tissues, we performed transcriptomic analyses on the SCAT and EWAT isolated from mock-treated and IAV-infected mice ($p ≤ 0.05$ and fold-change

(FC) cutoff of ≥2). In SCAT, 1214 genes were found differentially expressed between mock-treated and IAV-infected mice. In EWAT, 660 differentially expressed genes (DEGs) were identified. Venn diagrams were used to display comparison of the lists of upregulated or downregulated DEGs (Fig. 3a). Upon infection, 296 genes were upregulated in both SCAT and EWAT, 268 genes were upregulated only in SCAT, and 177 genes were upregulated only in EWAT. Regarding infection-associated downregulated genes, 148 genes were commonly downregulated in SCAT and EWAT, 502 genes were downregulated only in SCAT, and 39 were downregulated only in EWAT. We then conducted canonical pathway analyses of these DEG lists using Ingenuity Pathways Analysis (IPA). The interferon signaling pathway was identified as the top canonical pathway with the highest positive

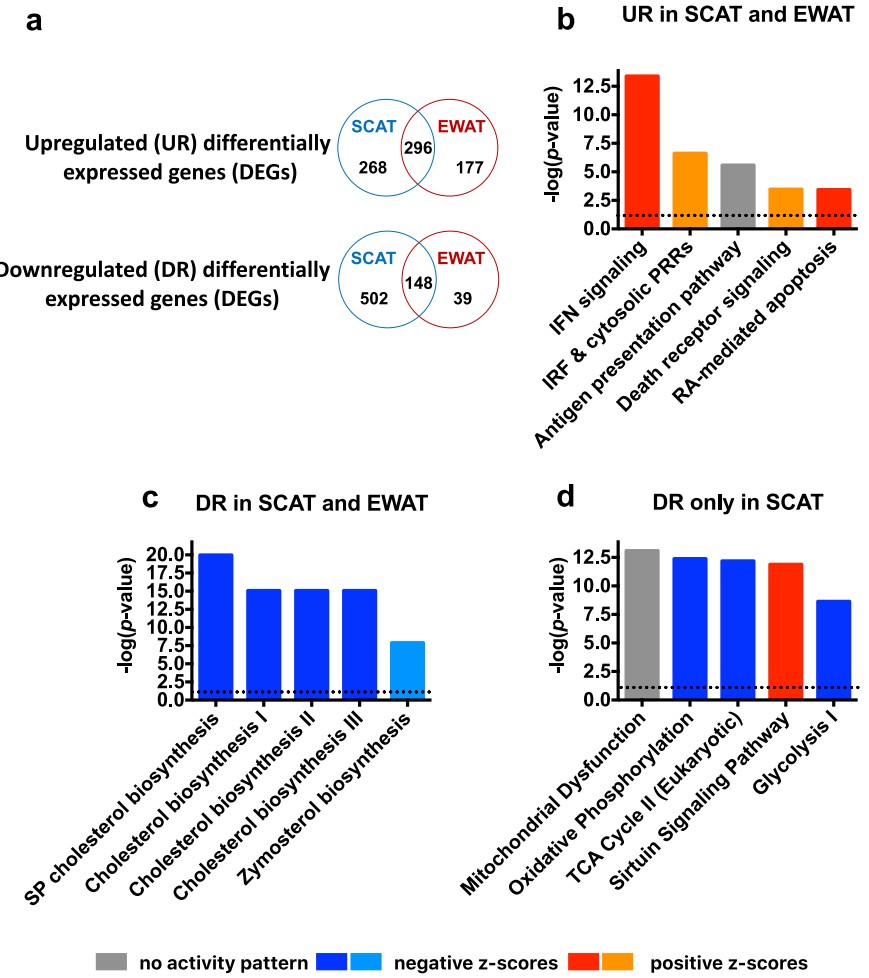

**Fig. 3 Transcriptomics shows fat-depot-specific response to influenza.** SCAT and EWAT from mock-treated and IAV-infected mice were treated for transcriptomic analysis at 7 dpi, $n = 4$ mock-treated animals and $n = 4$ IAV-infected animals. **a** Venn diagrams summarizing the number of distinct and overlapping differentially expressed genes (DEGs) in the SCAT and EWAT from IAV-infected mice vs. the SCAT and EWAT from mock-treated mice. Upregulated (UR) and downregulated (DR) DEGs numbers are shown. **b** The top five canonical pathways identified by Ingenuity Pathway Analysis (IPA) for the genes upregulated in SCAT and EWAT upon infection (IFN = interferon, IRF = interferon regulatory factor, RA = retinoic acid). The corresponding upstream regulators and their target molecules are shown in Supplementary Table 1. **c** The top five canonical pathways identified by IPA for the genes downregulated in SCAT and EWAT upon infection (SP = superpathway). The corresponding upstream regulators and their target molecules are shown in Supplementary Table 3. **d** The top five canonical pathways identified by IPA for the genes downregulated upon infection exclusively in SCAT (TCA cycle = tricarboxylic acid cycle). For (**b**) to (**d**), pathways were sorted by $p$ values and absolute z-scores. Intensities of red indicate the higher or lower value of positive z-scores (activated pathways), intensities of blue indicate the higher or lower value of negative z-scores (inhibited pathways), gray indicates pathways having no activity pattern available since no z-score could be calculated. The spotted line indicates IPA's default threshold.

z-score in the list of 296 genes upregulated in SCAT and EWAT, suggesting activation of the pathway in both fat depots during infection (Fig. 3b). For these upregulated genes, IPA predicted potential upstream regulators. The top activated upstream regulators and their targets are listed in Supplementary Table 1. Interestingly, the transcription factors IRF3 and IRF7 are recognized as key regulators of type I IFN gene expression induced by viruses[30].

IPA identification of pathways populated by genes upregulated during infection only in one type of fat depots showed that the Rho-GTPase family signaling pathways were activated in SCAT, and the T-cell-driven-immune/inflammatory pathways were activated in EWAT (yet with relatively modest $p$ values and percentages of overlap) (Supplementary Table 2).

From the core set of 148 genes that were downregulated during infection in both SCAT and EWAT, IPA ranked multiple pathways linked to cholesterol biosynthesis as being the most

significantly associated with infection (Fig. 3c). Concordantly, the top upstream regulators were predicted to be the transcription factors sterol regulatory element binding proteins (SREBPs, also called SREBFs) and the endoplasmic reticulum (ER) protein SCAP, all of which are master regulators of cholesterol biosynthesis[31] (Supplementary Table 3). Interestingly, transcriptomic data indicated the inhibition of oxidative phosphorylation (OXPHOS), the tricarboxylic acid (TCA) cycle, and glycolysis during infection, but only in SCAT (Fig. 3d). Indeed, within the mitochondrial energy metabolic pathways, numerous genes coding for functional/structural components of the electron transport-linked OXPHOS (predominantly in complex I) as well as TCA cycle genes were found to be downregulated in SCAT (Supplementary Fig. 3). Concomitantly, the sirtuin signaling pathway was activated in this fat depot (Fig. 3d). Sirtuins are a group of $NAD^+$-dependent protein-deacetylating stress-responsive enzymes that regulate glucose and lipid metabolism,

inflammation and WAT browning, the latter through deacetyla-tion regulation of pathways upstream of UCP1[32,33]. It is noteworthy that opposite regulation directions of the OXPHOS pathway and the sirtuin signaling pathway have been recently reported[34].

Overall, these results showed that influenza infection was associated with major transcriptional changes in fat tissues. In both SCAT and EWAT, interferon signaling pathways were activated and cholesterol biosynthesis pathways were repressed. Importantly, infection impacted major metabolic pathways (glycolysis, OXPHOS, TCA cycle) only in SCAT.

**Influenza durably alters the host's energy metabolism.** Adipose tissue is a key metabolic organ in the regulation of whole-body energy homeostasis[16,18]. Thus, we looked at whether influenza infection impacted on the host systemic metabolism. At 7 dpi, blood glucose levels were lower in IAV-infected mice than in mock-treated controls, whereas insulin levels were higher. Blood levels of resistin—originally described as a WAT-derived hor-mone modulating insulin resistance in rodents[35]—were lower in infected animals than in controls (Fig. 4a). Remarkably, these changes in blood glucose, insulin and resistin levels persisted long after the infection had resolved (up to 20 weeks post-infection) (Fig. 4b). However, these long-lasting metabolic sequelae had no impact on the animals' glucose tolerance, as revealed by an intraperitoneal glucose tolerance test (Supplementary Fig. 4a).

Studies in various rodent models have shown that white-to-brown adipose tissue remodeling prevents high-fat-diet (HFD)-induced obesity[27,36]. Hence, we hypothesized that IAV-infected mice would be protected against obesity. Mock-treated and IAV-infected mice were fed an HFD from 7 dpi (the time when SCAT browning became apparent). As expected, mock-treated mice fed an HFD (mock HFD) gained more weight than mock-treated mice fed a standard diet (mock SD). Although IAV-infected mice fed an HFD (IAV HFD) gained less weight than mock-treated mice fed an HFD (mock HFD), they still gained more weight than when fed a standard diet (IAV SD) (Fig. 4c, and Supplementary Fig. 4b for the corresponding area under the curve (AUC)). Accordingly, IAV-infected mice were not protected against HFD-induced glucose intolerance (Fig. 4d, and Supplementary Fig. 4c for the corresponding area under the curve (AUC)). However, it is noteworthy that HFD-fed IAV-infected animals showed lower levels of blood glucose and resistin than HFD-fed mock-treated mice (Fig. 4e), as was observed at 7 dpi and 20 weeks post-infection under standard diet conditions (see Fig. 4a, b).

These results indicated that influenza infection has a long-lasting impact on whole-body glucose metabolism that is independent of the host's dietary status (it occurred in lean and obese conditions).

**Viral protein-expressing cells are detected in adipose tissues.** Next, we sought to quantify viral RNA in the adipose tissues from IAV-infected mice (*M1* negative strand specific qPCR assay). Although viral RNA was detected at lower levels than in the lungs, it was detected in SCAT and EWAT (but not in pancreas) at 7 dpi (Fig. 5a). To get some insight into the dynamics of viral RNA accumulation, viral RNA levels were quantified in the lungs, SCAT and EWAT at 2, 4, 7, 14, and 28 dpi (Fig. 5b). In the lungs, the viral load rapidly increased from 2 dpi, peaked at 7 dpi, and rapidly declined thereafter. In SCAT and EWAT, we also observed a time-dependency of viral RNA accumulation, which started from 2 dpi, culminated at 7 dpi and decreased afterward, as noticed in the lungs.

WAT's cellularity is highly heterogenous[18–20,22]. Mature, lipid-filled adipocytes make up ~30% of adipose tissue cells. The

remaining stromal vascular fraction (SVF) cells include vascular cells, fibroblasts, innate and adaptive immune cells, and adipocyte precursor cells (preadipocytes). We therefore separated mature adipocytes from SVF cells through enzymatic fractionation of the SCAT and EWAT from mock-treated and IAV-infected mice (Fig. 5c). As shown in Supplementary Fig. 5a, viral RNA was detected in SVF cells isolated from SCAT and EWAT. No viral RNA was detected in adipocytes isolated from either fat depot. Thus, immune cells (identified as CD45$^+$ cells) and preadipocytes (identified as CD45$^-$ CD31$^-$ CD29$^+$ CD34$^+$ Sca-1$^+$ cells) from the SVF of the SCAT and EWAT from mock-treated and IAV-infected mice were sorted by fluorescent-activated cell sorting (FACS) (see gating strategy Supplementary Fig. 5b) and stained for the viral protein hemagglutinin (HA) (Fig. 5d). HA-positive immune cells were found in the SCAT of nearly (84%) all infected mice, a frequency that was much lower in the EWAT (17%). Half of infected mice had HA-positive preadipocytes in the SCAT, while none were detected in any of the EWAT samples.

Thus, immune cells and (less frequently) preadipocytes bearing viral RNA (M1) and viral protein (HA) were detected in the adipose tissues (mostly in SCAT) of influenza-infected mice. In contrast, no adipocytes bearing viral RNA were found in either fat depot.

**Influenza induces brown-like adipogenesis in preadipocytes.** Preadipocytes share numerous phenotypic and functional prop-erties with macrophages[37]. IAV replication in macrophages is most often abortive (no release of new infectious virions), yet certain virus strains can replicate productively in these cells[38,39]. Thus, the detection of HA-positive preadipocytes in the WAT from infected mice (see Fig. 5d) led us to question the IAV-infection ability of murine preadipocytes in vitro. We used mouse preadipocyte 3T3-L1 cells, which can differentiate into adipocytes upon induction with a standard hormone cocktail[40]. Hence, undifferentiated 3T3-L1 cells (from now on referred to as pre-adipocytes) were exposed to graded doses of IAV (at MOI of 0.01, 0.25, 0.5, 1, or 2) and viral RNA levels were evaluated at 6, 24, and 48 h post-infection (hpi). As shown in Fig. 6a, viral RNA levels rose from 6 to 24 hpi at each MOI; this suggested efficient IAV infection and viral genome replication in preadipocytes. It is noteworthy that infection did not affect the viability of pre-adipocytes, in contrast to highly permissive Madin-Darby canin kidney (MDCK) cells (Supplementary Fig. 6a).

Unlike most RNA viruses, IAVs transcribe and replicate their genome inside the nucleus; this induces major nuclear and nucleolar ultrastructural changes[41]. As shown in Fig. 6b, high-resolution transmission electron microscopy revealed the marked accumulation of viral M1-associated rod-like tubular structures in the nucleus, and the complete segregation of nucleolar components in infected preadipocytes (MOI of 5, at 24 hpi), relative to mock-treated cells. In addition, the particles budding from the preadipocytes' plasma membranes appeared to be empty (Fig. 6b). Consistently, no infectious particles were detected in the super-natants of infected preadipocytes while fully infectious virions were released from infected murine lung epithelial cells (MLE-15 cells) (MOI of 0.5, at 6, 24, 48, and 72 hpi) (Supplementary Fig. 6b).

We next sought to determine preadipocytes' response to infection (MOI of 1, at 24 hpi) by quantifying the expression of the antiviral innate-immune-related genes *Tlr3* (coding for Toll-like receptor 3), *RigI* (coding for retinoic acid-inducible protein I), *Mda5* (coding for melanoma-differentiation associated protein 5), *Mx1* (coding for Mx1), and *Vip* (coding for viperin) (Fig. 6c). Relative to mock-treated cells, the expression of these genes was enhanced in infected preadipocytes. It is worth mentioning that in vitro infection of preadipocytes at much lower MOI (0.01) also

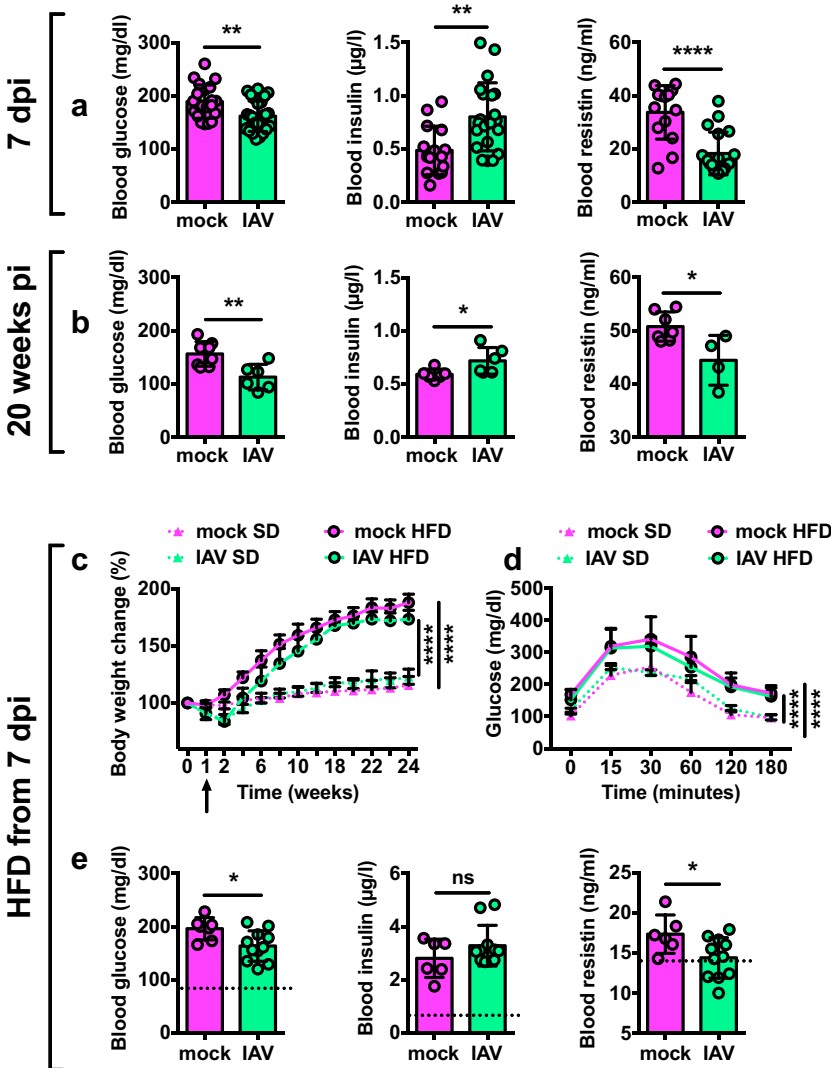

**Fig. 4 Influenza durably alters the host's energy metabolism. a** Blood levels of glucose (10–12 h fasting, mg/dl), insulin (10–12 h fasting, μg/l) and resistin (fed state, ng/ml) in mock-treated and IAV-infected mice at 7 dpi. For glucose: $n = 20$ mock-treated animals and $n = 30$ IAV-infected animals. For insulin and resistin: $n = 15$ mock-treated animals and $n = 15$ IAV-infected animals. Individuals values, as well as means ± SD are shown. **$p < 0.01$, ****$p < 0.0001$. **b** Blood levels of glucose, insulin and resistin in mock-treated and IAV-infected mice at week 20 post-infection, $n = 8$ mock-treated animals and $n = 6$ IAV-infected animals. Individuals values, as well as means ± SD are shown. *$p < 0.05$, **$p < 0.01$. **c** Body weight change (% initial body weight) of mock-treated and IAV-infected mice fed with standard diet (SD) or with high-fat diet (HFD) from 7 dpi (arrow), $n = 5$ mock-treated SD-fed animals (mock SD), $n = 10$ IAV-infected SD-fed animals (IAV SD), $n = 10$ mock-treated HFD-fed animals (mock HFD), and $n = 15$ IAV-infected HFD-fed animals (IAV HFD). Data are expressed as mean ± SD. ****$p < 0.0001$. Corresponding area under the curve (AUC) is shown in Supplementary Fig. 4b. **d** Intraperitoneal glucose tolerance test (IP-GTT) was performed in 10–12 h-fasted mice, 18 weeks post-infection. Blood samples for glucose level determination were taken from the tail vein before glucose administration (i.p. injection, 1 g/kg BW) and after 15, 30, 60, 120, and 180 min, $n = 5$ mock-treated SD-fed animals, $n = 10$ IAV-infected SD-fed animals, $n = 9$ mock-treated HFD-fed animals and $n = 12$ IAV-infected HFD-fed animals. Data are expressed as mean ± SD. ****$p < 0.0001$. Corresponding area under the curve (AUC) is shown in Supplementary Fig. 4c. **e** Blood levels of glucose, insulin and resistin (10–12 h fasting) at sacrifice, $n = 6$ mock-treated HFD-fed animals and $n = 11$ IAV-infected HFD-fed animals. The dotted lines indicate the values obtained for mock-treated SD-fed animals ($n = 4$). Individuals values, as well as means ± SD are shown. *$p < 0.05$, ns = not significant. Differences between mock-treated and IAV-infected groups were considered significant when $p < 0.05$.

led to increased *Tlr3*, *RigI*, *Mda5*, *Mx,1* and *Vip* transcript levels (Supplementary Fig. 6c).

Importantly, infection enhanced the expression of genes involved in the browning process, such as *Ucp1*, *Pgc1a* (coding for peroxisome proliferator-activated receptor gamma coactivator 1 alpha), *Fgf21* (coding for fibroblast growth factor 21), *Apln* (coding for apelin), *Metrnl* (coding for meteorin-like protein), and *Tmem26* (coding for transmembrane protein 26)[42–45] in preadipocytes (MOI of 1, at 24, 48, and 72 hpi) (Fig. 6d). Accordingly, adipocytes that had differentiated in vitro from

IAV-infected preadipocytes had higher *Ucp1* and *Pgc1a* transcript levels than adipocytes that had differentiated from mock-treated preadipocytes (Supplementary Fig. 6d).

Although no viral RNA-harboring adipocytes were detected in either fat depots from IAV-infected mice, previous literature reported that adipocytes might be permissive to IAV infection in vitro[46,47]. Thus, differentiated 3T3-L1 preadipocytes (from now on referred to as adipocytes) were infected with IAV, in vitro. As shown in Supplementary Fig. 7, infection had no impact on adipocytes' viability (Supplementary Fig. 7a), and

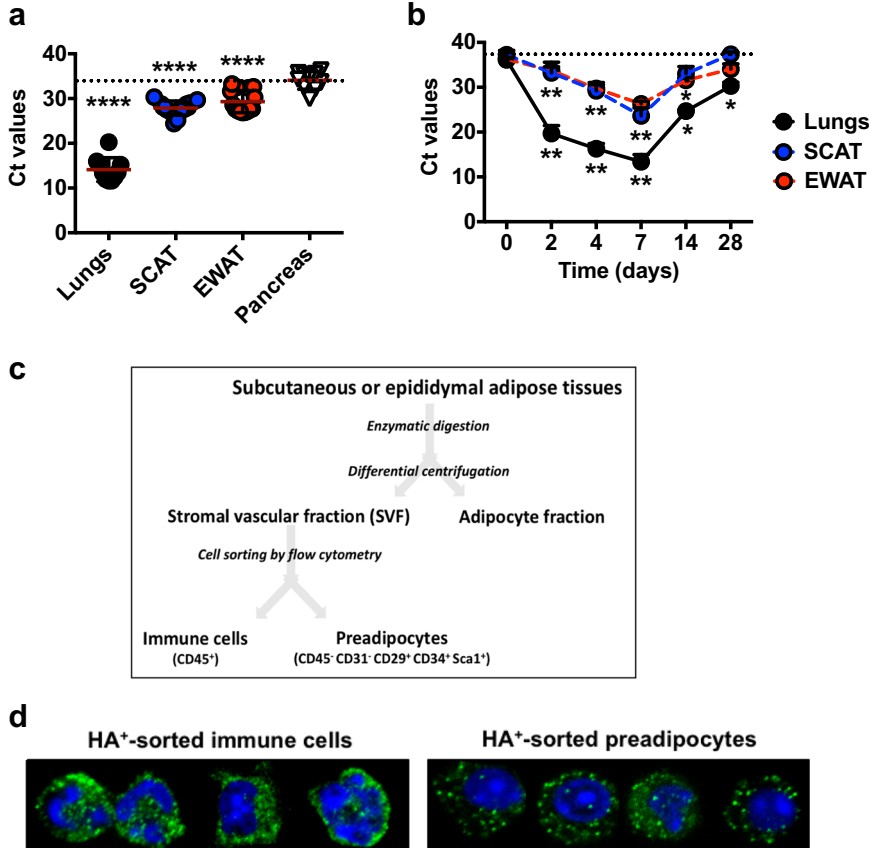

**Fig. 5 Viral protein-expressing cells are detected in adipose tissues. a** Viral RNA levels (expressed as cycle threshold (Ct) values) in the lungs, SCAT, EWAT, and pancreas at 7 dpi. Lungs and pancreas from $n = 10$ IAV-infected animals, and SCAT and EWAT from $n = 25$ IAV-infected animals. The dotted line indicates pancreas' mean Ct value. Individuals values, as well as means ± SD are shown. ****$p < 0.0001$. **b** Viral RNA levels (expressed as Ct values) in the lungs, SCAT, and EWAT from mock-treated and IAV-infected mice, at 0, 2, 4, 7, 14, and 28 dpi, $n = 5$ mock-treated animals and $n = 8$ IAV-infected animals per time point. The dotted line indicates day 0 of infection. Data are expressed as mean ± SD. *$p < 0.05$, **$p < 0.01$. **c** Fractionation procedure used to separate stromal vascular fraction (SVF) cells from mature adipocytes, of the adipose tissues. Immune cells (characterized as CD45+ cells) and preadipocytes (characterized as CD45− CD31− CD29+ CD34+ Sca-1+ cells) from the SVF were then sorted by flow cytometry (see gating strategy Supplementary Fig. 5b). **d** A selection of sorted immune cells and preadipocytes expressing the viral protein hemagglutinin (green) observed by immunofluorescence and confocal microscopy. The nuclear counterstain DAPI was used (blue). All images were taken at ×63 magnification, $n = 3$ mock-treated animals and $n = 6$ IAV-infected animals. Differences between pancreas and lungs, pancreas and SCAT and pancreas and EWAT (**a**) and between day 0 and days 2, 4, 7, 14, or 28 (**b**) were considered significant when $p < 0.05$.

resulted in increased levels of viral RNA (Supplementary Fig. 7b), nuclear and nucleolar changes featuring efficient viral genome replication, and budding of empty particles (Supplementary Fig. 7c), as well as increased expression of the antiviral innate-immune-related genes *Tlr3*, *RigI*, *Mda5*, *Mx1*, and *Vip* (Supplementary Fig. 7d). As opposed to preadipocytes (see Fig. 6d), IAV infection of adipocytes did not increase the expression of the browning-like/beiging-related genes *Ucp1*, *Pgc1a*, *Fgf21*, *Apln*, *Metrnl*, and *Tmem26* (Supplementary Fig. 7e).

Importantly, in vitro IAV infection of human primary preadipocytes (MOI of 1, 48 hpi) significantly increased the expression of the browning-like/beiging-related genes *Ucp1*, *Prdm16* (coding for PR domain containing 16), *Tmem26*, and *Tbx15* (coding for T-Box15). As observed with mouse cells, infection of human adipocytes (differentiated from human primary preadipocytes) did not increase the expression of these genes (with the exception of *Ucp1*) (Supplementary Fig. 7f).

These data showed that preadipocytes and adipocytes are permissive to in vitro IAV infection; however, virus replication is abortive. Most importantly, in vitro infection induced the commitment of preadipocytes to brown-like/beige adipocytes, and not the white-to-beige conversion of mature adipocytes.

**Influenza induces metabolic reprogramming of preadipocytes.** A comparative analysis of the transcriptomes of in vitro mock-treated and IAV-infected (MOI of 1, at 24 hpi) preadipocytes (3T3-L1 cells) and adipocytes (differentiated from 3T3-L1 cells) showed that more genes were up- or downregulated following infection in preadipocytes than in adipocytes (1472 genes vs. 585 genes, respectively), and that more genes were upregulated than downregulated in both cell types (981 upregulated genes vs. 491 downregulated genes in preadipocytes, and 566 upregulated genes vs. 19 downregulated genes in adipocytes) (Fig. 7a). IPA analyses revealed that the 449 genes that were upregulated upon infection both in preadipocytes and adipocytes were related to pathways endowed with interferon signaling, virus sensing and antiviral immune response (Supplementary Table 4).

The 117 genes that were only upregulated in adipocytes were related to stress-signaling pathways, yet with rather low $p$ values and overlap percentages (Supplementary Table 5). Strikingly, for the 532 genes exclusively upregulated in preadipocytes, the top-scored canonical pathways were the serine-glycine bio-synthesis, folate transformations (such as polyglutamylation), and glycine biosynthesis, pathways (Supplementary Table 6). These metabolic processes are connected by the serine-glycine-

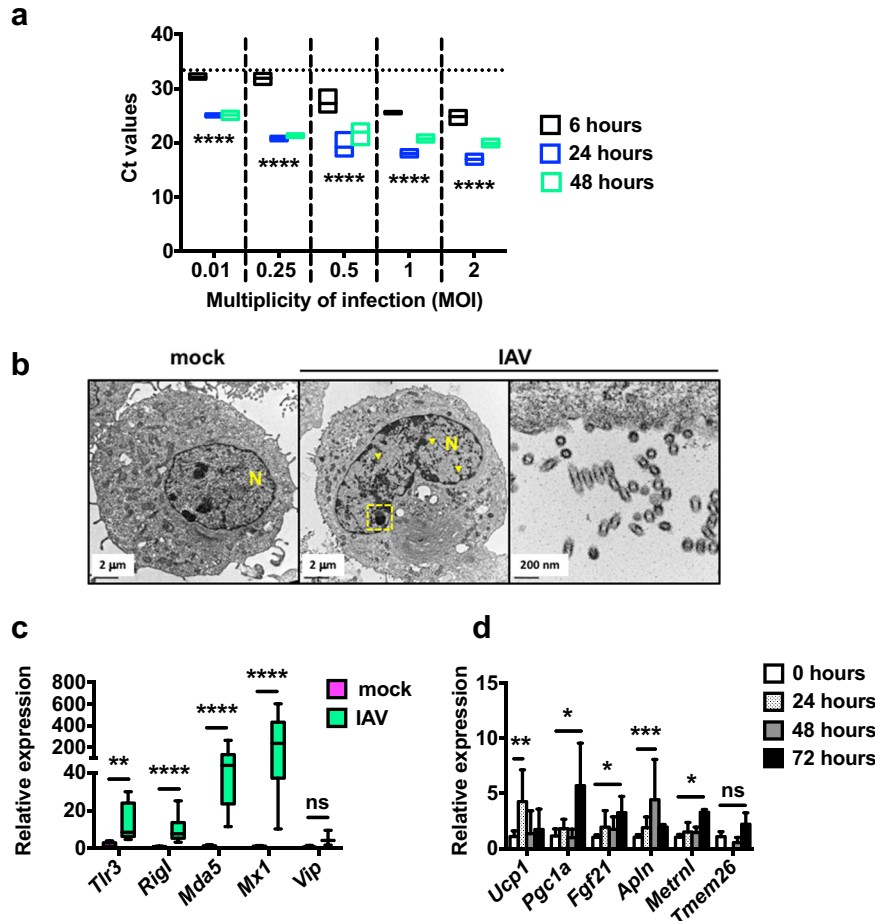

**Fig. 6 Influenza induces brown-like adipogenesis in preadipocytes. a** Viral RNA levels (expressed as Ct values) in IAV-infected preadipocytes (3T3-L1 cells) (MOI of 0.01, 0.25, 0.5, 1, or 2) at 6, 24, and 48 h post-infection (hpi), $n = 4$ biologically independent samples per condition. The dotted line indicates the 6 hpi/0.01 MOI mean Ct value. ****$p < 0.0001$. **b** Representative transmission electron microscopy sections of mock-treated and IAV-infected preadipocytes (MOI of 5, 24 hpi), $n = 3$ biologically independent samples per condition. Classical nuclear (yellow arrowheads) and nucleolar (yellow dashed frame) fingerprints associated with IAV infection are shown. A detailed view of empty viral particles at budding regions is presented. N = nucleus. **c** Relative mRNA expression (RT-qPCR) of the antiviral innate-immune-related genes *Tlr3*, *RigI*, *Mda5*, *Mx1*, and *Vip* in mock-treated and IAV-infected preadipocytes (MOI of 1, 24 hpi). For *Tlr3* and *Vip*: $n = 5$ biologically independent samples per condition. For *RigI*, *Mda5*, and *Mx1*: $n = 7$ biologically independent mock-treated preadipocyte samples and $n = 18$ biologically independent IAV-infected preadipocyte samples. **$p < 0.01$, ****$p < 0.0001$, ns = not significant. **d** Relative mRNA expression of the genes involved in the browning-like/beiging adipogenesis process *Ucp1*, *Pgc1a*, *Fgf21*, *Apln*, *Metrnl*, and *Tmem26* in mock-treated and IAV-infected preadipocytes at 24, 48, and 72 hpi (MOI of 1), $n = 6$ biologically independent samples per condition. Of note, for *Tmem26* expression, the 24 hpi time was not done. Data are expressed as mean ± SD. *$p < 0.05$, **$p < 0.01$, ***$p < 0.001$, ns = not significant. Differences between 6 and 24 hpi and between 6 and 48 hpi (**a**), and between mock-treated and IAV-infected groups (**c**, **d**) were considered significant when $p < 0.05$.

one-carbon (SGOC) pathway; an offshoot of glycolysis[48]. Indeed, the IAV-infection-gene-signature in preadipocytes includes several genes encoding enzymes within the SGOC pathway, such as the cytosolic phosphoglycerate dehydrogenase (PHGDH), phosphoserine phosphatase (PSPH) and serine hydroxymethyltransferase (SHMT) 1, and the mitochondrial SHMT2, methylenetetrahydrofolate dehydrogenase (NADP⁺ dependent) 1-like (MTHFD1L) and MTHFD2 (Fig. 7b). It is worth mentioning that the upregulation of several enzymes involved in SGOC metabolism has been recently reported for IAV-infected macrophages[49]. The upstream regulators predicted to be associated with the genes exclusively upregulated in preadipocytes included activating transcription factor 4 (ATF4), UCP1, and MYC (Supplementary Table 6). ATF4 and MYC form a positive feedback loop for transcriptional activation of the SGOC pathway[50], and UCP1 was reported to induce *Atf4* expression[51].

The SGOC metabolism generates diverse outputs, such as the synthesis of lipids, nucleotides, ATP, and proteins[48], suggesting that the infected preadipocytes' translational machinery might have been activated. Indeed, tRNA charging was also identified as a top-ranked canonical pathway activated in infected preadipocytes (Supplementary Table 6). Concordantly, the mRNA expression levels of genes related to ribosome biogenesis (*Urb2*, coding for unhealthy ribosome biogenesis protein 2, *Wdr75*, coding for WD repeat domain 75, *Zfp593*, coding for zinc finger protein 593, *Mrto4*, coding for MRT4 homolog, ribosome maturation factor and *Nop58*, coding for nucleolar protein 58) were enhanced upon in vitro infection in preadipocytes and not in adipocytes (Fig. 7c).

Only a small number of genes were downregulated in adipocytes alone ($n = 13$) or in both preadipocytes and adipocytes ($n = 6$); no biological functions or pathways could therefore be assigned. In preadipocytes, pathways related to

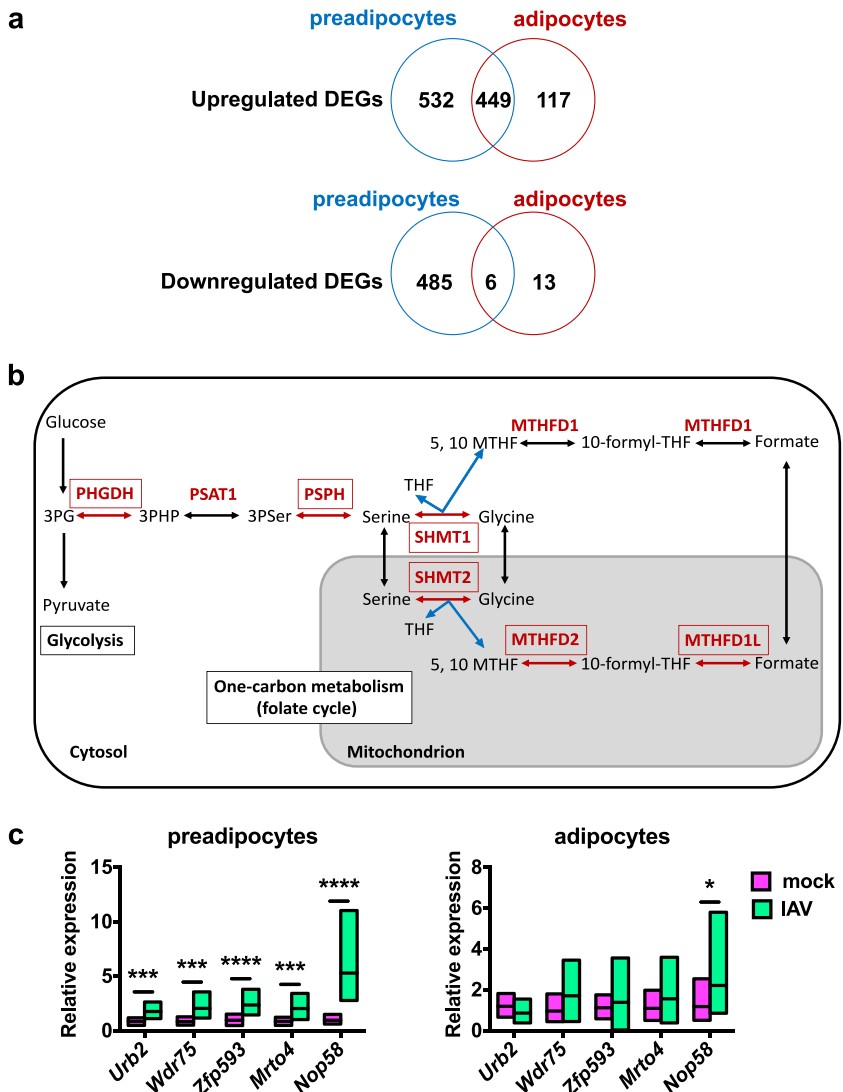

**Fig. 7 Influenza induces metabolic reprogramming of preadipocytes.** Murine preadipocytes (3T3-L1 cells) and adipocytes (differentiated from 3T3-L1 cells) were infected at a MOI of 1, and samples were prepared for transcriptomic analysis at 24 hpi, $n = 4$ biologically independent samples per condition. **a** Venn diagrams showing the numbers and intersections of the differentially expressed genes (DEGs), either upregulated or downregulated, in IAV-infected vs. mock-treated preadipocytes and adipocytes in the whole-genome expression profiling microarray data set. Top canonical pathways and related upstream regulators are listed in Supplementary Table 4 (genes commonly upregulated upon infection in preadipocytes and adipocytes), Supplementary Table 5 (genes only upregulated upon infection in adipocytes), Supplementary Table 6 (genes only upregulated upon infection in preadipocytes), and Supplementary Table 7 (genes only downregulated upon infection in preadipocytes). **b** Scheme of the serine-glycine-one-carbon (SGOC) metabolic pathway. Genes coding for cytosolic and mitochondrial SGOC enzymes (in red) that were upregulated in preadipocytes after IAV infection are framed. **c** Relative mRNA expression (RT-qPCR) of genes related to ribosome biogenesis (*Urb2*, *Wdr75*, *Zfp593*, *Mrto4*, and *Nop58*) in mock-treated and IAV-infected preadipocytes and adipocytes (MOI of 1, 24 hpi), $n = 12$ biologically independent samples per condition. Data are expressed as mean ± SD. *$p < 0.05$, ***$p < 0.001$, ****$p < 0.0001$. Differences between mock-treated and IAV-infected groups (**c**) were considered significant when $p < 0.05$.

amino acid catabolism and cholesterol biosynthesis were found to be downregulated upon infection, yet with modest $p$ values and overlap percentages (Supplementary Table 7).

Altogether, transcriptome analyses suggested activation of the interferon-dependent antiviral response in both preadipocytes and adipocytes upon in vitro infection, and illuminated intrinsic differences between the two types of cells. Indeed, preadipocytes dealt with infection by acquiring a metabolic program mainly characterized by the transcriptional activation of the SGOC metabolic pathway.

**Influenza changes mitochondrial bioenergetics of adipose cells.** It has been reported that IAV infection decreases host cell's respiration[52]. We above showed that in vitro IAV infection of

preadipocytes induced their commitment to brown-like/beige adipocytes, increased the expression of genes involved in the SGOC metabolic pathway, and was associated with the elevated expression of genes involved in ribosome biogenesis—all processes known to impact cellular bioenergetics[53,54]. Thus, we used high-resolution respirometry to record oxygen consumption rates (OCRs) of mock-treated vs. IAV-infected preadipocytes (3T3-L1) and adipocytes (differentiated from 3T3-L1) at 0, 24, 48, and 72 hpi (MOI of 1). The OCRs were measured in the resting respiratory state and after the sequential addition of specific mitochondrial inhibitors or uncouplers. These steps enable us to ascribe OCR values into distinct mitochondrial and cellular processes (proton ($H^+$) leakage, maximal respiratory state, and non-mitochondrial residual respiratory state) (Supplementary

Fig. 8a). As expected from the literature[55], all respiratory states were significantly higher in mock-treated adipocytes than in mock-treated preadipocytes (Supplementary Fig. 8b). As shown in Supplementary Fig. 8c, infection impacted bioenergetics differently in preadipocytes and adipocytes. In preadipocytes, IAV infection tended (as expected from the results presented above) to increase the OCRs after 48 h but this rise was not statistically significant. Conversely, the cellular respiration of adipocytes progressively decreased after IAV infection (as has been described for highly IAV-permissive MDCK cells[52]) and the difference reached significance at 72 hpi.

## Discussion

Persistent, chronic infections by viruses, parasites or bacteria often lead to metabolic anomalies due to the functionality impairment of metabolic tissues or organs such as the liver, skeletal muscle, pancreas, brain, and WAT[56]. Lymphocytic choriomeningitis virus, human and simian immunodeficiency viruses, adenovirus, vaccinia virus, and several species of parasites and bacteria have been reported to target murine and human WAT directly[57–61]. In contrast, it is not clear whether acute infections also result in long-lasting metabolic changes—notably by altering WAT's functions. This is a compelling question, given the frequency and diversity of the acute infections with which modern societies are burdened.

Here, we report that a single acute (intranasal) infection with a respiratory pathogen (seasonal influenza A virus subtype H3N2) profoundly modifies WAT's metabolic and immune functions, in mice. Transcriptome analyses of the subcutaneous and visceral fat depots in response to influenza infection (Fig. 8a) revealed the activation of type I interferon signaling pathways associated with the concomitant inhibition of cholesterol biosynthesis pathways both in SCAT and EWAT. The reciprocal negative cross-talk between antiviral type I IFN responses and cholesterol homeostasis is increasingly being recognized[62]. Reconfiguring cholesterol metabolism is now considered as an integral component of the host's response to infection both through heightening antiviral immunity[63,64], and creating an unfavorable environment for virus stability and infectivity[65]. Interestingly, the transcriptional response to infection also differed according to the localization of fat depots. Indeed, pathways related to energy metabolism were altered upon infection only in SCAT: the glycolysis, TCA cycle and OXPHOS metabolic pathways were repressed, while the sirtuin signaling pathway was activated. The mammalian sirtuin (SirT) family is categorized as stress-induced-NAD$^+$-dependent protein deacetylases that regulate processes such as energy metabolism and inflammation[32]. Several SirT family members have been reported to promote WAT lipolysis[66] and WAT browning[67], two processes that are closely linked[68]. In accordance with SirT signaling pathway activation, *Mgl* expression level was increased and brown-like/beige adipocytes were observed in the SCAT from infected mice; this suggested increased lipolysis[69] and thermogenesis. Indeed, expression of the thermogenic UCP1 protein was increased in SCAT upon infection, and this correlated with elevated surface temperature, as expected from literature[70].

In rodents, SCAT (less frequently EWAT) is readily able to convert to a brown-like state in response to environmental stimuli including chronic cold adaptation, exercise and nutritional challenges; as well as external and internal cues such as pharmacological treatment with β3-adrenergic receptor agonists and various peptides and hormones[71]. Leptin (which secretion is enhanced in the SCAT upon infection) and insulin (which blood level is increased in infected mice) have been reported to promote WAT browning[72]. Influenza infection may also have favored the

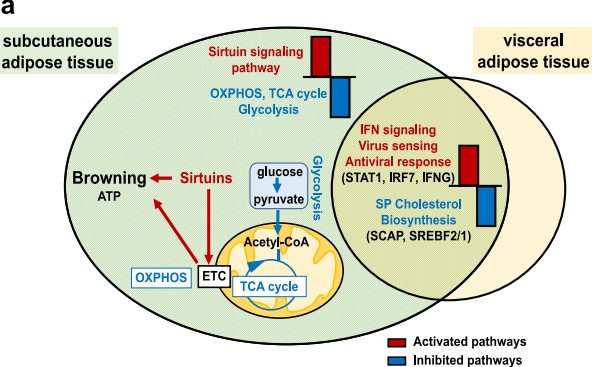

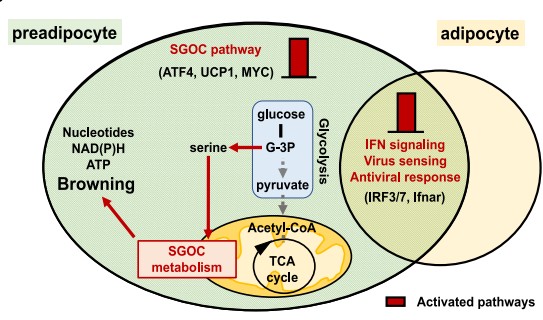

**Fig. 8 Metabolic reprogramming associated with influenza infection in subcutaneous adipose tissue (in vivo) and in preadipocytes (in vitro).** **a** Transcriptome analysis of the subcutaneous (SCAT) and the visceral (EWAT) adipose tissues from mock-treated and IAV-infected mice revealed activation of the STAT1-dependent-IFN-mediated signaling pathway and the concomitant inhibition of the cholesterol biosynthesis pathways in both SCAT and EWAT. IAV infection was associated with rewired energy metabolism only in the SCAT, as reflected by activation of the sirtuin signaling pathway and inhibition of glycolysis, OXPHOS, and TCA cycle. This was associated with a white-to-brown phenotypic change of the SCAT. **b** In vitro IAV infection of preadipocytes and adipocytes led to activation of the IFN signaling pathways in both cell types. In preadipocytes only, IAV infection rewired cell metabolism through activation of the SGOC metabolic pathway. This was associated with the induction of a brown-like/beige genetic program in preadipocytes. OXPHOS = oxidative phosphorylation, ETC = electron transport chain, SGOC metabolism = serine-glycine-one-carbon metabolism. Activated and inhibited pathways are indicated in red and blue, respectively. Top upstream regulators are indicated in brackets.

accumulation of immune cells associated with WAT browning, such as M2 macrophages, eosinophils or innate lymphoid type 2 cells[73], in the SCAT.

Strikingly, we found that immune cells harboring viral proteins (HA) were present in the WAT (mostly SCAT) of influenza-infected mice. Immune cell egress from the lungs has been reported in the context of influenza infection[74,75]. Thus, immune cells (either infected or harboring remnant viral antigens and/or RNA) may have entered the bloodstream through the capillaries of damaged alveolar wall of the infected lungs and reached fat depots, participating to the observed changes in WAT's secretory function and to the induction of antiviral innate-immunity-related pathways. The phenotypic characterization of immune cells that are present in the fat depots of IAV-infected mice is currently under investigation.

Alongside immune cells, viral antigen-positive preadipocytes were also found, although less frequently, in the WAT from infected mice. How preadipocytes acquired viral antigens from immune cells that have migrated from infected lungs to WAT,

remains to be established. Also, determining whether other WAT's cell types are impacted during influenza infection should be investigated since WAT's functionality result from the concerted interplay between all these cells. Single-cell RNA sequencing combined with flow cytometry or mass cytometry (CyTOF) could help answer this key question[76].

Using in vitro approaches (Fig. 8b), we showed that pre-adipocytes—that share many properties with macrophages[37]—are permissive to IAV infection, yet replication was abortive. Importantly, infection induced the browning-like/beiging program in preadipocytes and this was associated with the activation of the folate-dependent SGOC pathway, an offshoot of glycolysis[48] recently reported to drive effector responses to IAV infection in macrophages[49].

The impact of acute infections on the metabolism of the affected hosts has been rather set aside. Here, we reported a systemic response to influenza infection that modulates WAT's metabolic and immune functions. Importantly, infection was also associated with changes of host's metabolism that persist long after infection has been resolved. Altogether, these findings illustrate the complexity of influenza virus–host interactions, in which the adipose tissue may play an important role.

## Methods

**Mice, virus, and in vivo infection protocol.** Male C57BL/6JRj mice (6-week old; Janvier Labs, Le Genest-Saint-Isle, France) were housed and manipulated in an Animal Biosafety Level-2 facility, in strict accordance with Institut Pasteur's guidelines on animal care and use, and in compliance with European animal welfare regulations (European Communities Council Directive of 1986 revised in 2010, 2010/63/EU). Protocols were approved by the regional Animal Experimentation Ethics Committee (Comité d'Ethique en Expérimentation Animale, Hauts-de-France, CEEA 75) and the French Ministry of Higher Education and Research (Ministère de l'Enseignement Supérieur et de la Recherche) (authorization numbers: 00357.03 and 00033.02). Mice were fed ad libitum with low-fat diet (LFD: 10% kcal from fat) or, when mentioned, with high-fat diet (HFD: 60% kcal from fat) obtained from Research Diets (New Brunswick, NJ, USA). The human-derived, mouse-adapted influenza A/Scotland/20/1974 (H3N2) virus was grown, isolated, and titrated in MDCK cells.

Mice were administrated intranasally with a sublethal dose (30 PFU, in 50 μl PBS) of H3N2 (IAV groups) or with PBS (mock groups). Body weights were monitored. Intraperitoneal glucose tolerance tests and infrared thermography measurements were performed. Blood, lungs, subcutaneous (inguinal) and visceral (epididymal) white adipose tissue depots (SCAT and EWAT, respectively) were collected and weighed. SCAT and EWAT explants were cultured for 16–24 h for hormone and cytokine detection in culture supernatants by ELISA. White adipose tissues were processed for further histological, flow cytometry, western blot, or gene expression analyses.

**In vivo assessment of metabolic phenotyping.** For determination of intraperitoneal glucose tolerance, mice were fasted for 10–12 h before being injected (intraperitoneally) with 1 g/kg body weight of D-glucose (Sigma-Aldrich, Saint-Louis, Missouri, USA). Glucose levels were measured by tail-tip bleeding with an automatic glucometer (ACCU-CHEK Performa, Roche, Mannheim, Germany) before injection, and 15, 30, 60, 120, and 180 min after glucose administration.

For infrared thermography, an infrared camera (FLiR systems) was used to acquire static images of the SCAT region of unrestrained, anesthetized mock-treated, and IAV-infected mice, 7 dpi, n = 3 mock-treated animals and n = 4 IAV-infected animals (three measurements per mouse), as described in Crane et al.[77].

**Ex vivo mouse adipose tissue explant cultures.** SCAT and EWAT samples were isolated from mock-treated and IAV-infected mice (7 dpi), and cultured as described in Thalmann et al.[25]. Tissue explants were put into 6-well plates (~50 mg/well) (Sarstedt, Nümbrecht, Germany) and incubated for 24 h in DMEM without serum and supplemented with 1% penicillin/streptomycin (Life Technologies, CA, USA) in a humidified 37 °C, 5% CO2 incubator. After the incubation period, the medium was aspired from the floating fat and centrifuge for 5 min at 500 × g at room temperature to remove any excess lipids or debris. The resultant explant culture supernatants were stored at −80 °C until further analysis.

**Mouse adipose tissue histology and histomorphometry.** SCAT and EWAT were harvested from mock-treated and IAV-infected mice, at 7 dpi. After overnight fixation in 4% paraformaldehyde, tissue samples were embedded in paraffin. Multiple sections (5 μm) were de-paraffinized, rehydrated, and stained with hematoxylin and eosin (H&E). For evaluation of adipocyte number and size by

morphometry, at least 10 fields per H&E-stained slide were visualized and images were digitally captured using a digital camera (AxioCam HRc, Zeiss, Göttingen, Germany) connected to an optical microscope (Axioplan 2 Imaging, Zeiss). Adipocyte area (in μm²) was measured using an in-house macro specifically developed for automated image analysis of WAT on the Fiji-ImageJ software (NIH).

**Cell fractionation of mouse adipose tissue.** Individual SCAT and EWAT were carefully excised from mock-treated and IAV-infected mice at 7 dpi. Tissues were thoroughly minced with scissors (1–2 mm³ pieces) and digested in collagenase I (1 mg/ml, Sigma-Aldrich) for 1 h at 37 °C with gentle shaking by inversion every 20 min. Digested tissues were then passed through a 250 μm nylon filter and centrifuged at 150 g for 17 min at room temperature (RT). The floating fraction—containing intact, purified mature adipocytes with minimal stromal cell contamination (see Supplementary Fig. 5c)—was carefully recovered with a wide opening plastic transfer pipette and processed for further gene expression analyses. Remaining infranatants were centrifuged at 400 × g for 5 min, RT. The erythrocytes from the SVF cell pellets were lysed with cold lysis buffer (3 min, on ice) and SVF cells were filtered through a 90 μm and then a 40 μm nylon filter before being washed twice by centrifugation at 400 × g for 5 min, RT. SVF cells were counted and resuspended in PBS 0.5% BSA for further flow cytometry sorting.

**Identification and sorting of cells from mouse adipose tissue.** SVF cell suspensions were first incubated with an Fc blocking reagent (anti-mouse CD16/CD32, clone 2.4G2, BD Pharmingen, San Jose, CA, USA). Afterward, cells were stained with anti-mouse CD31 (BV421, clone 390), CD45 (BV510, clone 30-F11), CD34 (PE, clone HM34), CD29 (AF700, clone HMβ1-1) from BioLegend, and Sca-1 (APC-Cy7, clone D7) from eBioscience, at a dilution of 1:100. Samples were then directly run into the BD Influx™ cell sorter (BD Biosciences) equipped with a 86-μm nozzle and tuned at a pressure of 24.6 psi and a frequency of 48.25 kHz. Sample fluid pressure was adjusted to reach a throughput rate of 10,000 events per second. Immune cells and preadipocytes were selected as CD45+ and CD45− CD31− CD34+ CD29+ Sca-1+, respectively. The gating strategy is shown on Supplementary Fig. 5b.

**In vitro differentiation of preadipocytes and infection procedure.** The mouse 3T3-L1 preadipose cell-line (ATCC®, CL-173™) was used. 3T3-L1 cells were cultured in Dulbecco's modified Eagle's medium (DMEM), high glucose, Gluta-MAX™ (Gibco, ThermoFisher Scientific, Waltham, MA, USA) supplemented with 10% heat-inactivated fetal bovine serum (FBS) (Gibco) and 1% penicillin/streptomycin in a humidified 37 °C, 5% CO2 incubator. The differentiation of 3T3-L1 cells into mature adipocytes was conducted as indicated in ref. [40]: 3T3-L1 pre-adipose cells were seeded in 6-well plates (Sarstedt) at a density of 4 × 10⁵ cells per well, in 2 ml of culture medium. Two days after reaching confluence, cells were incubated in the differentiation medium containing 1 μg/ml insulin (Sigma-Aldrich, Saint-Louis, Missouri, USA), 0.25 μM dexamethasone (Sigma-Aldrich) and 0.5 mM 3-isobutyl-1-methylxanthine (IBMX) (Sigma-Aldrich) for 2 days. The medium was then replaced by the maintenance medium that consists of DMEM supplemented with 10% FBS, 1% penicillin/streptomycin and 1 μg/ml insulin. The maintenance medium was renewed every 2 or 3 days.

Human primary preadipocytes were purchased from Lonza (Basel, Switzerland) together with the differentiation kit and the culture medium. Preadipocytes were induced to differentiate into fully mature adipocytes according to manufacturer's instructions.

Mouse or human preadipocytes and adipocytes (4 × 10⁵) were infected with IAV at 0.01, 0.25, 0.5, 1, 2, or 5 multiplicities of infection (MOIs), as indicated. Mock-treated cells were used as controls. Cell viability was determined using the Orangu™ colorimetric assay (Cell Guidance Systems, Cambridge, UK). Supernatants were collected for quantification of released infectious particles by using the plaque-forming assay on MDCK cells. Cells were used for western blot, transmission electron microscopy, immunostaining, or gene expression analyses.

**High-resolution respirometry on mouse adipose cells.** Oxygen consumption rates (OCRs) of IAV-infected (MOI of 1) and mock-treated preadipocytes and adipocytes (10⁶ cells), were determined using high-resolution respirometry (Oxy-graph-2k, OROBOROS Instruments, Innsbruck, Austria) at 24, 48, and 72 h post-infection (hpi). Experiments were performed under controlled temperature (37 °C) and stirring (750 rpm).

OCRs were measured under baseline conditions (recording resting respiration) and in response to, successively, 2 μg/ml oligomycin (ATP synthase inhibitor; recording proton leak respiration, H+ Leak), pulses of 0.5 μM carbonyl cyanide m-chlorophenylhydrazone (CCCP) (mitochondrial uncoupler; recording maximal respiration) and 2.5 μM antimycin A (mitochondrial chain complex III inhibitor; recording residual respiration). Data acquisition and analyses were performed with DatLab 4 software (OROBOROS Instruments).

**Western blotting.** Tissues (lungs, SCAT and EWAT) were homogenized in lysis buffer supplemented with protease inhibitors (cOmplete™ Protease Inhibitor Cocktail, Sigma-Aldrich). Whole tissue protein extracts (40–50 μg) were

fractionated by 10% SDS-PAGE and transferred onto polyvinylidene difluoride membranes using a transfer apparatus as per the manufacturer's instructions (Life Technologies). Membranes were blocked with 5% bovine serum albumin (BSA) in PBS/0.1% Tween for 1 h, washed and probed overnight at 4 °C with polyclonal antibody against UCP1 (1:500) (Abcam, Cambridge, UK). Membranes were washed and incubated with horseradish peroxidase-conjugated goat anti-rabbit antibody (KPL, Milford, MA, USA) for 2 h. After three washings, blots were developed with the ECL system (Bio-Rad Laboratories) according to the provided protocol. GAPDH (Santa Cruz Biotechnology, Dallas, TX, USA) was used as protein loading control.

**Transmission electron microscopy**. In vitro mock-treated and IAV-infected (MOI of 5) mouse preadipocytes and adipocytes ($5 \times 10^6$) were fixed in 4% glutaraldehyde for 15 min at 4 °C and post-fixed in cacodylate (0.2 M, pH 7.4)/4% glutaraldehyde (1:1) for 30 min at 4 °C. Cell pellets were embedded in EPON™ resin (Sigma-Aldrich) that was allowed to polymerize at 60 °C for 48 h. Ultrathin sections were cut, stained with 5% uranyl acetate and 5% citrate and deposited onto gold electron microscopy grids for examination using a JEOL 1011 transmission electron microscope (JEOL Ltd., Tokyo, Japan).

**Immunofluorescence using confocal microscopy**. Immune cells and pre-adipocytes sorted from the SCAT and EWAT of mock-treated and IAV-infected mice (7 dpi) were fixed in 4% paraformaldehyde for 15 min at room temperature (RT), carefully washed with PBS/0.1% Tween-20, and permeabilized with 0.1% Triton X-100. After a blocking step with PBS/1% BSA (1 h, RT), cells were over-night (4 °C) labeled with the anti-H3N2 hemagglutinin (HA) polyclonal antibody (ThermoFisher Scientific, Waltham, MA, USA) before being incubated (1 h, RT) with the secondary antibody (goat anti-rabbit Alexa Fluor® 488-labeled, Invitrogen), at concentration recommended by the suppliers. Nuclei were then stained with DAPI (ThermoFisher Scientific) and coverslips and slides were assembled using a mounting medium (ProLong® Gold Antifade Mountant, ThermoFisher Scientific). Preparations were observed on a confocal laser microscope (Zeiss LSM 880, Oberkochen, Germany).

**Gene expression quantification by real-time RT-PCR**. Total RNAs were extracted from snap-frozen tissues (SCAT, EWAT and lungs), from the stromal vascular and adipocyte fractions isolated from adipose tissues, or from in vitro cultured murine or human preadipocytes and adipocytes, according to the protocol supplied with the RNeasy Lipid Tissue mini-kit (Qiagen, Hilden, Germany).

For RT-PCR amplification of all genes except the gene encoding the viral protein M1, RT was performed with the High-capacity RNA-to-cDNA kit (Applied Biosystems, Foster City, CA, USA) with oligo(dT)$_{16}$ primers and PCR was performed with the Power SYBR Green PCR Master Mix (Applied Biosystems) using the QuantStudio™ 12K Flex Real-Time PCR System (Applied Biosystems). Reactions were run in duplicate and the geometric mean of the housekeeping gene *Eef2* transcript levels was used as internal control to normalize the variability in expression levels. Expression data were analyzed using the $2^{-\Delta\Delta Ct}$ method.

For quantification of viral RNA, a negative strand specific RT-qPCR assay for IAV RNA encoding the *M1* gene (segment 7) was performed. Total RNAs were treated with RNAse OUT (Invitrogen) and reverse-transcribed with SuperScript® II Reverse Transcriptase (Invitrogen) using primer specific for *M1* (5′-TCT AAC CGA GGT CGA AAC GTA-3′). PCR was performed with TaqMan Universal PCR Master Mix (Applied Biosystems), detection primer pairs for *M1* (sense: 5′-AAG ACC AAT CCT GTC ACC TCT GA-3′; anti-sense: 5′-CAA AGC GTC TAC GCT GCA GTC C-3′) and *M1* specific TaqMan probe ((FAM) 5′-TTT GTG TTC ACG CTC ACC GTG CC-3′ (TAMRA)), using the QuantStudio™ 12K Flex Real-Time PCR System (Applied Biosystems).

**Whole-genome transcriptomics of adipose cells and tissues**. To prevent DNA contamination, RNAs extracted from adipose cells and tissues were treated with DNAse. The isolated RNAs were then dissolved in RNAse-free water, after which the concentration of total RNAs was measured and the quality was assessed for further microarray expression analyses. The transcriptional pro-filings of mock-treated vs. IAV-infected adipose cells and tissues were per-formed using Agilent's SurePrint G3 Mouse Gene Expression v2 8x60K Microarray kits (Agilent Technologies, Santa Clara, CA, USA), according to manufacturer's instructions.

The microarray analysis of in vitro mock-treated and IAV-infected mouse preadipocytes and adipocytes (MOI of 1, 24 hpi) was performed at the Centre de Ressources Biologiques pour la Génomique des Animaux Domestiques et d'Intérêt Economique (CRB GADIE), INRA Jouy-en-Josas Research Center, France. Separate microarrays were run for each experimental sample (one sample per culture condition, four independent experiments). A single-color design was used to provide two types of comparisons: [mock-treated preadipocytes vs. IAV-infected preadipocytes] and [mock-treated mature adipocytes vs. IAV-infected mature adipocytes]. Identification of differentially expressed genes and functional investigation were done using GeneSpring software (Version 14.8, Agilent Technologies). Differentially expressed genes were identified using a moderated T test. A Benjamini–Hochberg false discover rate (FDR) was then used as multiple

testing correction method and a corrected *p*-value cutoff of 1% was applied. A fold-change >2 cutoff was then added to select for the differentially expressed genes between the mock and infected conditions.

The microarray analysis of adipose tissues (SCAT and EWAT) of mock-treated and IAV-infected mice (30 PFU, 7 dpi) was performed at the Institut Pasteur's platform of Transcriptomics and Applied Genomics. Separate microarrays were run for each experimental sample (two experimental groups, four mice per group, two tissues per mouse). After hybridization, slides were scanned on a SureScan Microarray scanner (Agilent Technologies) and further processed using Feature extraction v10.7.3.1 software. The resulting text files were uploaded into language R v3.5.1 for LIMMA (Linear Model for Microarray Data) differential expression analysis. A within-array normalization was performed using LOWESS (locally weighed linear regression) to correct for dye and spatial effects. Moderated T-statistic with empirical Bayes shrinkage of the standard errors was then used to determine significantly modulated genes. Statistics were corrected for multiple testing using a FDR approach.

For functional analyses, Gene Ontology categories (www.geneontology.org) and Ingenuity Pathway Analyses (IPA, www.qiagen.com/ingenuity, Ingenuity Systems, Inc., Redwood City, CA, USA) were performed on the lists of commonly or exclusively differentially up- or downregulated genes.

**ELISA**. Insulin levels were quantified using an Ultrasensitive Mouse Insulin ELISA (Mercodia, Uppsala, Sweden). Leptin and resistin levels were measured using: Mouse/Rat Leptin and Mouse Resistin Quantikine ELISA kits (R&D Systems, Minneapolis, MN, USA). IL-1β and IL-6 dosages were carried out with ELISA Ready-SET-Go kits (eBioscience/Affymetrix, ThermoFisher Scientific). IL-10 levels were determined using BD OptEIA™ kits (BD Biosciences, Franklin Lakes, NJ, USA).

**Statistics and reproducibility**. Differences between experimental groups were determined using unpaired two-tailed student's *t* test or analysis of variance (ANOVA) as appropriate and were considered significant when *p* values were lower than 0.05. For repeated measures (body weight evolution, IP-GTT tests), two-way ANOVA analyses with Bonferroni post-test were performed. For respirometry analyses, a one-way ANOVA followed by Tukey post hoc test was carried out when comparing four groups or more. Statistics were performed using GraphPad Prism 6 software.

**Reporting summary**. Further information on research design is available in the Nature Research Reporting Summary linked to this article.

## Data availability
Minimum information about Microarray Experiment (MIAME) on in vitro mock-treated vs. IAV-infected preadipocytes, and in vitro mock-treated vs. IAV-infected adipocytes, was deposited in ArrayExpress database (https://www.ebi.ac.uk/arrayexpress/) under the platform ID E-MTAB-6646.

Minimum information about Microarray Experiment (MIAME) on the subcutaneous (inguinal) adipose tissue (SCAT) of mock-treated vs. IAV-infected mice, and the visceral (epididymal) adipose tissue (EWAT) of mock-treated vs. IAV-infected mice, was deposited in ArrayExpress database (https://www.ebi.ac.uk/arrayexpress/) under the platform ID E-MTAB-6646.

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

## Acknowledgements
We appreciate valuable discussions with Drs. Kassem Makki and Sandra Weller, and advice and support from Drs. Jean-Claude Sirard and Jean Dubuisson, and from Sia Praline and Nor Snø. We thank Dr. Corinne Grangette for providing reagents needed for mouse preadipocyte and adipocyte culture and differentiation, Dr. Odile Poulain-Godefroy for the kind gift of human primary Preadipocytes, and Marie-Josée Ghoris for technical help. Jérôme Lecardonnel (INRA, UMR GABI, Jouy-en-Josas, France) and Marie-Hélène Gevaert (Laboratoire d'Histologie, Faculté de Médecine, Lille, France) are acknowledged for their expert technical assistance in, respectively, microarray hybridization (adipose cells) and WAT histology. We also thank Denis Ressnikoff and Elisabeth Erazzuriz (Centre Imagerie Quantitative Lyon Est (CIQLE), Université Lyon 1, Lyon, France) for their technical assistance in confocal and transmission electron microscopy. We thank Antonino Bongiovanni and Sophie Salomé-Desnoulez as well as Drs. Elizabeth Werkmeister, Nicolas Barois and Hélène Bauderlique of the BioImaging Center Lille (BICeL, Lille, France) for access to systems and expert advises on microscopy and flow cytometry. Dr. Cécile Lecoeur and Peggy Bouquet (Transcriptomics and Applied Genomics Group, Lille) are thanked for, respectively, advice on transcriptomic analyses, and technical assistance in microarray hybridization (adipose tissues). Institut Pasteur's animal facility staff is also thanked for its assistance. The CPER-Région Hauts-de-France's financial contribution to the acquisition of the Oroboros O2k-respirometer is acknowledged. This study was supported by the Centre National de la Recherche Scientifique (CNRS) and the Institut National de la Santé et de la Recherche Médicale (INSERM). AA received a grant from Lille University. I.W. and F.T. received a grant from CNRS. The funders had no role in study design, data collection and analyses, decision to publish, or preparation of the paper.

## Author contributions
A.A., S.L., J.B., A.P., A.M.-L., M.B., L.D., D.S., T.J., O.M.-C., L.S., and V.S. performed the experiments. A.A., M.R.-C., S.L., A.M.-L., M.B., D.H., C.F., O.M.-C., D.D., R.L.G., F.T., and I.W. planned the experiments and/or analyzed the data, I.W., A.A., and F.T. wrote the paper, and I.W. oversaw the project. All authors provided critical feedback and approved the content and the submission of the paper.

## Competing interests
The authors declare no competing interests.
