## [Peer Review File · Communications Biology]

Editorial Note: *This manuscript has been previously reviewed at another Nature Research journal. This document only contains reviewer comments and rebuttal letters for versions considered at Communications Biology. Mentions of the other journal have been redacted.*

Reviewers' comments:

Reviewer #1 (Remarks to the Author):

The paper advances our understanding of the ramifications of influenza (IAV) infection biology and maybe influential to those studying co-morbidity of IAV and obesity, an important topic. Regarding the scientific quality and technical soundness of the work, apart from an inappropriate extrapolation (below LOD) and the lack of some controls I am confident in the quality. The impression the manuscript gives is that the authors are trying to fit these data to claims they are invested in (i.e. IAV infected adipocytes directly). However, these data are compelling irrespective of this claim, a claim that is not substantiated and even appears to be disproved. Likewise claims of IAV changes metabolism in cell intrinsic, extrinsic, and systemically are unfounded and unnecessary. The authors need to adjust the claims and text as recommended repeatedly by referees.

Communications Biology review of Manuscript: [REDACTED]

Communications Biology main paper criteria:

Novelty- moderate

The paper does not provide strong evidence for its conclusions

Most of the data are technically sound.

The manuscript is of value to researchers in IAV and obesity.

Brief summary of the manuscript: IAV infects adipocytes, changes their metabolism, "browns them" and this results in systemic metabolic consequences for the host long after the infection is cleared.

Overall impression of the work- The paper advances our understanding of the ramifications of influenza (IAV) infection biology and maybe influential to those studying co-morbidity of IAV and obesity, an important topic. Regarding the scientific quality and technical soundness of the work, apart from an inappropriate extrapolation (below LOD) and the lack of some controls I am confident in the quality. The impression the manuscript gives is that the authors are trying to fit these data to claims they are invested in (i.e. IAV infected adipocytes directly). However, these data are compelling irrespective of this claim, a claim that is not substantiated and even appears to be disproved. Likewise claims of IAV changes metabolism in cell intrinsic, extrinsic, and systemically are unfounded and unnecessary.

Specific comments, with recommendations for addressing each comment

Previous referees asked the title and abstract to be softened, I would add the text to this request. A great many claims are not substantiated by the experiments, and adjectives that related statistical significance are not used appropriately. Previous referees have brought up important points that the authors overlook (below), the authors need to address these, and this makes subsequent reviewers like myself less enthusiastic about putting in the time and effort. Nonetheless, I am excited enough about this work to express my concerns, reiterate concerns of previous referees, and implore the authors to be satisfied with summarizing their actual data and presenting it in a way that bolsters the readers confidence in these important experiments.

Referee 1 asks for the in vitro assays to be performed Ex Vivo with tissue in Q5. The authors then remove adipose and attempt in vitro infections. Obviously, this was destined to fail and not what was meant by "should be performed directly ex vivo." Referee1 is concerned, as am I, about immune cell infiltrates in the tissue as requests RNA-seq to distinguish their transcriptomes from the adipose. The authors respond with another analysis of the bulk transcriptomics already presented and claim to now "thoroughly analyze it." One would hope a thorough analysis would be presented in the first place, but this fails to address the reviewers concerns and may actually undermine the authors claims. Further, Referee 2 rightly wanted tissue staining and the authors respond with "Instead, we performed IAV-specific staining of preadipocytes and immune cells freshly isolated from the fat depots of infected mice." The authors should clarify as all of the figures I found were not from fat deposits of infected mice, they were "primary preadipocytes and adipocytes freshly isolated from the adipose tissue of mock-treated mice...infected in vitro." IF the authors are referring to Fig 5c, as outlined below this figure demonstrates the levels in AF are essentially zero and below the level of detection.

Referee 2 summarizes my concerns and that of referee 1 well, briefly no evidence is provided to eliminate trafficking viral antigens and RNA in immune cells or account for their influence, and the latter is repeatedly conflated with IAV direct effects. The authors do not present data to support the claim that influenza "directly" triggers the changes claimed especially given the in vitro data. Rather than removing over interpretation and reach the authors double down "Therefore, we believe that infected immune cells may egress from the infected lungs (peaking at 7 dpi) and, through lymphatic and/or blood vessels, reached fat depots. If, once in the adipose tissue, an infected immune cells is in the vicinity of adipose cells, the latter would be infected, which will affect their secretions and functions, leading to the observed changes at whole tissue levels." Notions like these are unsubstantiated by these data. An infiltrating immune cell is more likely to secrete mediators than transfer live infectious particles to the adipose. There is a wealth of literature demonstrating activated immune cells produce copious amounts of cytokines that alter metabolism (PMID: 30249998) and adipose (PMID: 27120716 PMID: 26330344). Importantly some viral infected localize to BAT (PMID: 31092580).

Major claims of the paper

Claim 1-Adipose tissues and cells (i.e. preadipocytes and adipocytes) may be previously unrecognized targets for influenza virus in vivo

- a. WAT contains IAV infected pre- & adipocytes-
- b. IAV cannot replicate in adipose (in vitro)

Fig 5b- Reviewers 1 & 2 previously pointed out my major concern the authors are not measuring IAV in adipocytes. The actual measurement appears to be of viral RNA and antigens being trafficked into the adipose tissue by infiltrating immune cells. It seems an unlikely coincidence that the kinetics match perfectly. The authors are harvesting fat tissue from the intraperitoneal area where these immune infiltrates are present and increase on the same time and amplitude as presented in Fig 5b. No attempt has been made or indicated to remove infiltrating immune cells from the tissue. OMICS of mixed cell populations should be referred to as such and not presented as if pure SCAT or EWAT was studied. Further, IAV has been shown to induce hemorrhages and hematomas in adipose tissue (DOI: 10.1128/JVI.01235-07) which as reviewer 1 pointed out only high path IAV would only transverse via blood in viremia but that doesn't exclude immune cells arriving via this route (which is supported by some of these data). Either way, these data indicate immune cells are present in these tissues and must be acknowledge in the text and taken into account in the interpretation of these data. IT IS NOT APPROPRIATE to refer to these cells as "cells in a Depot" or in any other ambiguous terms when the authors have defined them. Again, reviewers 1 (Q3) & 2 (Q1) are correct, these are extremely low levels of virus, especially for lungs during this time, ignoring the peak is delayed. In their rebuttal the authors argue "the kinetic (days 0, 2, 4, 7, 14 and 28 post-infection) of viral RNA detection in the fat depots that has been done in the revised version of the manuscript clearly show a time-dependency of viral RNA accumulation in fat tissues (Figure 5b). Importantly, the profiles in SCAT and EWAT were similar to that observed in the lungs, with a sharp peak at day 7 post-infection." This is troubling. The method of quantification the authors are using reports their values as RNA copies per cell with a limit of detection at 10^3 copies and assumes a cell contains 10 pg of RNA (PMID: 21185869). Fig 5b peak lung is approximately 10^7 copies per ug RNA. Applying Kawakami et al assumptions that roughly translate into 100 copies of M gene RNA per cell in the lung at peak. That is unbelievably low considering typical IAV lung titer is peaking at about 7 log₁₀ TCID₅₀ per ml lung homogenate. Further the authors point to the time delay as being specific to IAV accumulation in adipose, yet they appear to show a time delay in the lungs that is contrary to IAV kinetics in the lung. This, and the HA expression, supports the view that the authors are measuring viral antigen or RNA from immune cells. The authors present STAT1 and IFN antiviral defense as evidence for direct viral infection, which is untrue given the presence of virus harboring activated immune cells.

Fig 5c attempts to address the first issue by presorting the tissue and the results are clear, the adipocytes have no appreciable virus (i.e. AF are below the LOD, AF is not defined I assume adipose). That is probably why we see no effort to determine the significance. However, it is disconcerting that the reported adipose value is below 50 M1 copies/ugRNA even though the authors clearly state this value is far below the standard curve they generated to extrapolate their data See 715-716: "draw a standard curve (Ct value/viral copy number) ranging from 3.8 x10⁹ (Ct ~5) to 380 copies (Ct ~25); allowing virus quantification in samples by interpolation of the Ct values." Thus, this figure meaningless.

One assumes this extrapolation is well below the limit of detection elsewhere (PMID: 21185869) where quantification via this method uses a standard curve but the atypical unit copies per ugRNA makes it hard to compare to other methods papers that use copies per cell. Perhaps the authors used an unorthodox normalization that diminished their results (?). The PCR reaction conditions should be reported with cycles as this impacts results and threshold cycle (Ct) values of samples considered positive.

Overall, the Authors' responses gloss over and appears to ignore the elephant in the room that both reviewers are pointing out. It is somewhat alarming that in response to reviewer 2 Q1 that authors repeat the above arguments, then state "iv) despite the low RNA levels that were detected in SCAT and EWAT, transcriptome analysis showed significant induction of STAT1-dependentytype I IFN-mediated antiviral immune defense in both depots, and iv) HA-expressing immune cells and HA-expressing adipose cells were detected in the adipose tissues of infected animals." The later argument explains why cytokines are elevated here (claimed in the paper to be adipose secreted without support), how antigen got into adipose cells the authors can't infected, and why the adipose has induction of antiviral immune factors. These factors are well documented to alter metabolism in all cells, including adipose that could easily explain the "browning" effect.

TCID₅₀ would be helpful, but Supplementary Figure 6 is a-d, there is no "e" so I can't comment on resolution of the issues related to adipose infectivity in terms of kinetics and MOI. However, the copy levels appear within the LOD.

Figure 5c clearly demonstrates the immune cell containing SVF fraction is where the viral RNA are. Thus, Lines like 246 must be corrected as they are misleading. "Viral RNA was detected in the SCAT and EWAT (but not in 247 pancreas)" or 266 "Since virus-bearing adipose cells were detected in the adipose tissues of IAV-infected mice (see Fig. 5c and 5d)".

"IAC cannot replicate in adipose (in vitro)" implies it is infecting the adipose, and as Referee 1 points out, the cells are not being acid washed after in vitro infection so Fig 6a is not convincing as it reflects IAV adherence.

Fig 6c lacks uninfected controls to demonstrate background binding of NS1, NP, and HA antibodies so it should be removed and not discussed. (line 303-305) One wonders what the number of HA positive cells were in the mock (ref line 311). Statements like this are not substantiated without proper controls: "On the basis of this dataset, we conclude that influenza virus enters, replicates its genome, and expresses its viral proteins, in preadipocytes and adipocytes." Line 336

Claim 2- IAV may promote white adipose tissue browning by inducing preadipocytes to commit to brown-like/beige adipocytes.

ii. IAV induces thermogenic brown-like/beige adipocytes in SCAT

To claim this expression of UPC1 should be quantified. This is shown in Fig 2b-c. However, the error bars are expressed in SEM and do not reflect the variability of the representative blot. Change to Std Dev, note exact significance test performed, and change to scatter plot to represent the number of blots and mice used in graphical representation. Further, claims of "increased thermogenic profile" are not corroborated by the unknown statistical test and are indicated as the "most notable" evidence lines 166-170.

ii. "Figure 7b shows that infection led to elevated expression of browning related genes in preadipocytes but - strikingly - not in adipocytes." How do the authors reconcile the kinetics of IAV and preadipocytes turning brown to mediate the effects seen in the in vivo figures given this result?

Claim 3- IAV infection (of the host) leads to profound changes in the major WAT depots (i.e. the inguinal SCAT and the visceral (epididymal) adipose tissue (EWAT))

Fig 1b-Mass supports claim, fig should be scatter plot use Std dev & indicate statistical test

Fig1c- MGL deletion has no effect on adipose browning or brown adipose tissue activation (PMID: 21454566).

The claim that "We next used tissue explant cultures to investigate the impact of influenza infection on SCAT and EWAT's secretory functions" is misleading. These cytokines are systemically increased with IAV infection and the authors are not measuring secretion SCAT or EWAT. The authors also claim "leptin...was elevated in SCAT and low in EWAT" then "Importantly, IAV infection led to depot-specific differences in the adipose tissue's lipid metabolism, and leptin secretion.." IF the authors are trying to use Fig1d to bolster their argument that SCAT and EWAT are different then they should graph them together and use the appropriate statistical method to test this.

Fig 2 & 2 sup are at the heart of the claim that IAV promotes subcutaneous adipose tissue browning and is specific to SCAT not EWAT. Yet the histology slides were not stained for IAV and there is no indication as to blinding of the authors during data collection. This paper would benefit immeasurably from sending these slides out for independent analysis by a blinded histopathologist. If Supplemental 2a-b is a summary from a blinded expert, then this addresses some essential claims of the paper. These data should be presented in a meaningful way and the appropriate statistical analysis should be used to determine if IAV induces differences in browning of the SCAT and/or EWAT. The difference between IAV induced browning of SCAT and EWAT should be presented, I see no IAV induced difference between adipocyte area (i.e. In 2a the difference between SCAT and EWAT is constant for all 5 size groups irrespective of IAV). This should be moved to main figures and supported by raw numbers and representative histology.

All lines that state things like: "From the core set of the 148 genes that were commonly downregulated upon infection in SCAT and EWAT, "lines 194-195. This is one example of many, all should be changed to accurately reflect the data presented in the paper, namely IAV infection causes these local and systemic changes as opposed to IAV infection of SCAT or EWAT. Further, as the previous reviewers point out "OMICS" of these tissues include activated immune cells.

Claim 4- IAV infection rewires energy metabolism (transcriptomics): As outlined above the transcriptomics data is on a mixed cell population and infiltrating immune cells secrete cytokines that can alter adipose metabolism (PMID: 17118983; PMID: 30249998). This claim is valid as it refers to IAV infection in general and does not attempt to claim IAV infection of adipose cells alter their metabolism. Correct all text accordingly, except in the discussion where qualified reasonable claims may be made.

i. IAV rewires mitochondrial bioenergetics (cell intrinsic)- Cell intrinsic changes in bioenergetics between SCAT and EWAT were not measured or supported by these data and can't be claimed. The authors could easily perform bioenergetics on *in vitro* infections of pure cultures.

ii. IAV rewires mitochondrial bioenergetics (cell extrinsic at SCAT & EWAT tissue level)- Bioenergetics of adipose *ex vivo* is simple and would substantiate these claims (i.e. mice infected and tissue removed (days PI equal to transcriptomics) and subjected to mitochondrial stress tests). Alternatively, a relative comparison between IAV SCAT and EWAT transcriptomes could answer this question normalized by housekeeping gene. As it stands the authors are presenting only downregulated SCAT transcriptomic data (Fig 3d) without showing EWAT. Additionally, these "more through" analysis are presented incoherently. The authors present only one table allowing readers to compare SCAT & EWAT Table S1 and it is of transcripts upregulated with IAV yet the majority of the text refers to a figure without an EWAT comparison and infers extensively about TCA cycle and OXPHOS without measuring it.

*no explanation of DEG in text

Claim 5- "novel aspects of the metabolic control of influenza virus-host interactions at the molecular, cellular and tissue levels."- Claims of "influenza-associated modifications in SCAT and EWAT resulted in changes in the host's metabolism." are overreaching. The infection is systemic and has well documented systemic changes in glucose/insulin etc, that are referred to here as "metabolic." The persistence long after infection is indicative of these systemic changes. The authors could specifically infect adipose *in vivo* and repeat or soften claims and make distinctions between their metabolic claims in terms of host-tissue- SCAT/EWAT-cellular.

6. Minor claims

i. "influenza infection has a long-lasting impact on whole-body glucose metabolism, and that the impact is independent of the host's dietary status." Fig 4c doesn't show the IAV standard diet as indicated on line 235 and this figure and text is cumbersome due to the authors use of "mock" to indicate uninfected and uninfected standard diet.

ii. IAV changes host's glucose metabolism that persist long after infection has been cleared and IAV alters whole-body glucose metabolism long after resolution (>2wk). This has previously been reported, here is one example: PMID: 30203067, please refer to human studies too.

Are the claims novel? There are previous publications so lines 103-104 are not accurate, as was previously pointed out by referee 1, and should be changed to accurately reflect previous publications and claims of novelty including adipocytes as "previously unrecognized targets for influenza" should be stricken. Further, WAT browning has been seen with other viral infections so lines 477-8 need changed.

I refer the authors to HIV, LCMV, and CMV to soften their claims and bolster some of their arguments (PMID: 31110314; PMID: 26402858; PMID: 17118983; PMID: 30970254; PMID: 31092580; PMID: 26756119) and the numerous work in IAV & obesity.

General Considerations

The paper be of interest to others in the field and could be marginally influential.

Are the claims convincing? No. As they stand many of the claims lack evidence. As detailed above proper controls are missing, statistical analysis is either lacking or undefined, necessary comparisons to support claims are missing, and data undermining claims are present.

Are there other experiments that would strengthen the paper further? Ex vivo and in vitro bioenergetics would improve metabolic claims for extrinsic metabolic changes in tissue and SCAT vs EWAT respectively. They are easy and would dramatically support unsubstantiated claims from transcriptomics. Previous reviewers requested RNA-Seq & Ex Vivo tissue measure that were not performed. Quantitation of existing data could also help. These would be necessary to support the current claims of the paper. However, I recommend adjusting claims to support the current data.

Are the claims appropriately discussed in the context of previous literature? No, there is a substantial amount of literature in this area that is not cited enough.

If the manuscript is unacceptable in its present form, does the study seem sufficiently promising that the authors should be encouraged to consider a resubmission in the future? Yes.

Is the manuscript clearly written? No, the manuscript has several figures that provide little if any evidence and the presentation of the transcriptomics doesn't aid the reader in assessing the claims. It should be shortened and focused on claims based on these data and only relevant figures used.

Have the authors done themselves justice without overselling their claims? No, the authors have actually undermined their findings by overreaching on nearly every major claim and putting discussion worthy adjectives/descriptive details in results. These data are compelling even if adipocytes are not directly infected.

Statistical analysis- Unknown how sound, the statistical section of the methods makes it hard to determine what tests were applied to these data and legends lack test information.

Reproducibility- Methods are adequate unless indicated

COMMSBIO-19-1556A_AYARI et al.**Responses to Reviewer's comments and suggestions**

First, we would like to thank the Reviewer for the constructive criticisms, comments and suggestions. After completion of the suggested edits, the revised manuscript has significantly gained in precision and clarity.

All changes that have been made to fulfill Reviewer's requests are highlighted in the revised version of the manuscript.

In its revised form, the manuscript contains 8 Figures, 8 Supplementary Figures, 7 Supplementary Tables, and 77 references.

Major claims of the paper

Claim 1-Adipose tissues and cells (i.e. preadipocytes and adipocytes) may be previously unrecognized targets for influenza virus in vivo a. WAT contains IAV infected pre- & adipocytes- b. IAV cannot replicate in adipose (in vitro).

*** The first Reviewer's concerns regarding claim 1 is:** a. WAT contains IAV infected pre- & adipocytes (Fig. 5).

While the Reviewer acknowledged the presence of viral-antigen-positive preadipocytes in the fat tissues of influenza-infected mice (yet at lower magnitude than viral-antigen-positive immune cells), clarification for the detection of viral RNA in adipocytes was requested.

As suggested, the limit of detection of our qPCR assay has been added to all Figures that illustrate viral RNA quantification in tissues (SCAT, EWAT, lungs, pancreas), fat tissue-cell-fractions (stromal vascular cells vs. adipocytes), or cells (preadipocytes vs. adipocytes).

We agree with the fact that we could not write that adipocytes present in the fat tissues of influenza-infected mice were bearing viral RNA. This was inappropriate and, in fact, untrue. We thank the Reviewer for this insightful remark.

Thus, the Results concerning the search for the (possible) presence of viral RNA-, viral antigen- harboring cells in the fat tissues of influenza-infected mice, has been rewritten as follows:

Viral RNA- and viral antigen-harboring cells are detected in the adipose tissue of influenza-infected mice. Next, we sought to quantify viral RNA in the adipose tissues from IAV-infected mice (M1 negative strand specific qPCR assay). Albeit at lower levels than in the lungs, viral RNA was detected in SCAT and EWAT (but not in pancreas) at 7 dpi (Fig. 5a). To get some insight into the dynamics of viral RNA accumulation, viral RNA levels were quantified in the lungs, SCAT and EWAT at 2, 4, 7, 14 and 28 dpi (Fig. 5b). In the lungs, the viral load rapidly increased from 2 dpi, peaked at 7 dpi, and rapidly declined thereafter. In

SCAT and EWAT, we also observed a time-dependency of viral RNA accumulation, which started from 2 dpi, culminated at 7 dpi and decreased afterwards, as noticed in the lungs. WAT's cellularity is highly heterogenous^{18-20,22}. Mature, lipid-filled adipocytes make up approximately 30% of adipose tissue cells. The remaining stromal vascular fraction (SVF) cells include e.g. vascular cells, fibroblasts, innate and adaptive immune cells, and preadipocytes. We therefore separated mature adipocytes from SVF cells through enzymatic fractionation of the SCAT and EWAT from mock-treated and IAV-infected mice. As shown in Supplementary Fig. 5a, viral RNA was detected in SVF cells isolated from SCAT and not in SVF cells isolated from EWAT. No viral RNA was detected in adipocytes isolated from either fat depot (not shown). Thus, immune cells (identified as CD45⁺ cells) and preadipocytes (identified as CD45⁻ CD31⁻ CD29⁺ CD34⁺ Sca1⁺ cells) from the SVF of the SCAT and EWAT from mock-treated and IAV-infected mice were sorted by fluorescent-activated cell sorting (FACS) (see gating strategy Supplementary Fig. 5b) and stained for the viral protein hemagglutinin (HA) (Fig. 5d). HA-positive immune cells were found in the SCAT of nearly (84%) all infected mice, a frequency that was much lower in the EWAT (17%). Half of infected mice had HA-positive preadipocytes in the SCAT, while none were detected in any of the EWAT samples.

Thus, immune cells and (less frequently) preadipocytes bearing viral RNA (M1) and viral protein (HA) were detected in the adipose tissue (mostly in SCAT) of influenza-infected mice. In contrast, no adipocytes bearing viral RNA were found in either fat depot.

Corresponding to the revised Figure 5 and Supplementary Figure 5:

*** The second Reviewer's concerns regarding claim 1 is:** b. IAV cannot replicate in adipose (in vitro) (Fig. 6).

As suggested, illustration of the detection of the viral proteins NP, NS1 and HA viral in *in vitro* infected preadipocytes and adipocytes (immunofluorescent staining, and confocal microscopy) has been removed in the revised version of the manuscript.

In fact, the description of *in vitro* infection of preadipocytes and adipocytes has been completely reorganized, and shortened, as follows:

In vitro influenza virus infection is abortive in preadipocytes and triggers a brown-like/beige adipogenic program. Preadipocytes share numerous phenotypic and functional properties with macrophages³⁷. IAV replication in macrophages is most often abortive (i.e. no release of new infectious virions), yet certain virus strains can replicate productively in these cells^{38,39}. Thus, the detection of HA-positive preadipocytes in the WAT from infected mice (see Fig. 5d) led us to question the IAV-infection ability of murine preadipocytes in vitro. We used mouse preadipocyte 3T3-L1 cells, which can differentiate into adipocytes upon induction with a standard hormone cocktail⁴⁰. Hence, undifferentiated 3T3-L1 cells (from now on referred to as preadipocytes) were exposed to graded doses of IAV (at MOI of 0.01, 0.25, 0.5, 1 or 2) and viral RNA levels were evaluated at 6, 24 and 48 hpi. As shown in Fig. 6a, viral RNA levels rose from 6 to 24 hpi at each MOI; this suggested efficient IAV infection and viral genome replication in preadipocytes. It is noteworthy that infection did not affect the viability of preadipocytes, in contrast to highly permissive Madin-Darby Canin Kidney (MDCK) cells (Supplementary Fig. 6a).

Unlike most RNA viruses, IAVs transcribe and replicate their genome inside the nucleus; this induces major nuclear and nucleolar ultrastructural changes⁴¹. As shown in Fig. 6b, high-resolution transmission electron microscopy revealed the marked accumulation of viral M1-associated rod-like tubular structures in the nucleus, and the complete segregation of nucleolar components in infected preadipocytes (MOI of 5, at 24 hpi), relative to non-infected cells. In addition, the particles budding from the preadipocytes' plasma membranes appeared to be empty (Fig. 6b, right panel). Consistently, no infectious particles were detected in the supernatants of infected preadipocytes while fully infectious virions were released from infected murine lung epithelial cells (MLE-15 cells) (MOI of 0.5, at 6, 24, 48 and 72 hpi) (Supplementary Fig. 6b).

*We next sought to determine preadipocytes' response to infection (MOI of 1, at 24 hpi) by quantifying the expression of the antiviral innate-immune-related genes *Tlr3*, *RigI*, *Mda5*, *Mx1* and *Vip* (Fig. 6c). Relative to mock-treated cells, the expression of these genes was enhanced in infected preadipocytes. It is worth mentioning that *in vitro* infection of preadipocytes at much lower MOI (0.01) also led to increased *Tlr3*, *RigI*, *Mda5*, *Mx1* and *Vip* transcript levels (Supplementary Fig. 6c).*

*Importantly, infection enhanced the expression of genes involved in the browning process, such as *Ucp1*, *Pgc1a* (coding for peroxisome proliferator-activated receptor gamma coactivator 1 alpha), *Fgf21* (coding for fibroblast growth factor 21), *Apln* (coding for apelin), *Metnl* (coding for meteorin-like protein) and *Tmem26* (coding for transmembrane protein 26)⁴²⁻⁴⁵ in preadipocytes (MOI of 1, at 24, 48 and 72 hpi) (Figure 6d). Accordingly, adipocytes that had differentiated in vitro from IAV-infected preadipocytes had higher *Ucp1* and *Pgc1a* transcript levels than adipocytes that had differentiated from mock-treated preadipocytes (Supplementary Fig. 6d).*

Corresponding to the following revised Figure 6 and Supplementary Figure 6:

In addition, the transition for justifying *in vitro* infection of adipocytes (in the Results section), has been changed, as follows:

Although no viral RNA-harboring adipocytes were detected in either fat depots from IAV-infected mice, previous literature reported that adipocytes might be permissive to IAV infection *in vitro*^{46,47}. Thus, differentiated 3T3-L1 preadipocytes (from now on referred to as adipocytes) were infected with IAV, *in vitro*. Infection had no impact on adipocytes' viability (not shown). As shown in Supplementary Fig. 7, IAV infection of adipocytes resulted in increased levels of viral RNA (Supplementary Fig. 7a), nuclear and nucleolar changes featuring efficient viral genome replication, and budding of empty particles (Supplementary Fig. 7b), as well as increased expression of the antiviral innate-immune-related genes *Tlr3*, *RigI*, *Mda5*, *Mx1* and *Vip* (Supplementary Fig. 7c). As opposed to preadipocytes (see Fig. 6d), IAV infection of adipocytes did not increase the expression of *Ucp1*, *Pgc1a*, *Fgf21*, *Apln*, *Metnl* and *Tmem26* (Supplementary Fig. 7d). Importantly, *in vitro* IAV infection of human primary preadipocytes (MOI of 1, 48 hpi) significantly increased the expression of the browning-like/beiging-related genes *Ucp1*, *Prdm16* (coding for PR domain containing 16), *Tmem26* and *Tbx15* (coding for T-Box15). As observed with mouse cells, infection of human adipocytes (differentiated from human primary preadipocytes) did not increase the expression of these genes (with the exception of *Ucp1*) (Supplementary Fig. 7e).

Corresponding to the following revised Supplementary Figure 7:

Legend to Supplementary Fig. 7 a Viral RNA levels (expressed as Log₁₀ M1 RNA copies/µg RNA) in IAV-infected adipocytes (differentiated from 3T3-L1 preadipocytes) (MOI of 0.01, 0.25, 0.5, 1 or 2) at 6, 24 and 48 hpi, n = 4 to 6 per condition. The red marker indicates the virus input at 0 hpi. **b** Representative transmission electron microscopy sections of mock-treated and IAV-infected adipocytes (differentiated from 3T3-L1 preadipocytes) (MOI of 5, 24 hpi). Classical nuclear (yellow arrowheads) and nucleolar (yellow dashed frames) fingerprints associated with IAV infection are shown. On the right panel, a detailed view of empty viral particles at budding regions is presented. N = nucleus, LD = lipid droplet. **c** Relative mRNA expression (RT-qPCR) of the antiviral innate-immune-related genes *Tlr3*, *Rlg1*, *Mda5*, *Mx1* and *Vip* in mock-treated and IAV-infected adipocytes (differentiated from 3T3-L1 preadipocytes) (MOI of 1, 24 hpi), n = 5-7 mock and n = 5-18 IAV. **p<0.01, ****p<0.0001. **d** Relative mRNA expression of the genes involved in the browning-like/beiging adipogenesis process *Ucp1*, *Pgc1a*, *Fgf21*, *Apln*, *Metrn1* and *Tmem26* in mock-treated and IAV-infected adipocytes (differentiated from 3T3-L1 preadipocytes) at 24, 48 and 72 hpi (MOI of 1). Data are expressed as mean ± SD. **e** Relative mRNA expression of genes involved in the browning-like/beiging process (*Ucp1*, *Prdm16*, *Tmem26* and *Tbx15*) in mock-treated and IAV-infected primary human preadipocytes (left), and in mock-treated and IAV-infected primary human adipocytes (right), n = 10 mock-treated human preadipocytes, n = 6 IAV-infected human preadipocytes, n = 5 mock-treated human adipocytes and n = 6 IAV-infected human adipocytes. *p<0.05, **p<0.01, ***p<0.001. Differences between mock-treated and IAV-infected groups (**c**, **e**) were considered significant when p<0.05.

Claim 2- i. IAV may promote white adipose tissue browning by inducing preadipocytes to commit to brown-like/beige adipocytes. ii. IAV induces thermogenic brown-like/beige adipocytes in SCAT

*** The first Reviewer's concerns regarding claim 2 is:** i. IAV may promote white adipose tissue browning by inducing preadipocytes to commit to brown-like/beige adipocytes (Discussion).

We took into account the different remarks made by the Reviewer (such as infiltration of immune cells egressing from the infected lungs) to rewrite the Discussion on how influenza infection may lead to SCAT browning.

The text (Discussion) has been changed as follows:

In rodents, SCAT (less frequently EWAT) is readily able to convert to a brown-like state in response to environmental stimuli including chronic cold adaptation, exercise and nutritional challenges; as well as external and internal cues such as pharmacological treatment with α 3-adrenergic receptor agonists and various peptides and hormones⁷¹. Leptin (which secretion is enhanced in the SCAT upon infection) and insulin (which blood level is increased in infected mice) have been reported to promote WAT browning⁷². Influenza infection may also have favored the accumulation of immune cells associated with WAT browning, such as M2 macrophages, eosinophils or innate lymphoid type 2 cells⁷³, in the SCAT.

Strikingly, we found that immune cells harboring viral proteins (HA) were present in the WAT (mostly SCAT) of influenza-infected mice. Immune cell egress from the lungs has been reported in the context of influenza infection^{74,75}. Thus, immune cells (either infected or harboring remnant viral antigens and RNA) may have entered the bloodstream through the capillaries of damaged alveolar wall of the infected lungs and reached fat depots, participating to the observed changes in WAT's secretory function and to the induction of antiviral-innate-immunity-related pathways. The phenotypic characterization of immune cells that are present in the fat depots of IAV-infected mice is currently under investigation. Alongside immune cells, viral antigen-positive preadipocytes were also found, although less frequently, in the WAT from infected mice. How preadipocytes acquired viral antigens from immune cells that have migrated from infected lungs to WAT, remains to be established.

*** The second Reviewer's concerns regarding claim 2 is:** ii. IAV induces thermogenic brown-like/beige adipocytes in SCAT (Fig. 2a, 2b (histology and morphometry), and Fig. 2c, 2d (UCP1)).

- Fig. 2a, 2b. (histology & histomorphometry of SCAT from mock-treated mice vs. SCAT from IAV-infected mice).

Histology of the fat tissues has been made elsewhere than in our laboratory (see Acknowledgement section). For determination of adipocyte-size in the fat depots, this was done using an automatic program that measure the adipocyte size of ~ 1000 adipocytes per slide.

Crude data of the morphometry analysis have been provided as Supplementary data (4 Excel files):

- (1) Crude data morphometry analysis SCAT from mock-treated mice (values for each individual mouse (#8 to 14), and means),
- (2) Crude data morphometry analysis SCAT from IAV-infected mice (values for each individual mouse (#1 to 7), and means),
- (3) Crude data morphometry analysis EWAT from mock-treated mice (values for each individual mouse (#8 to 14), and means), and
- (4) Crude data morphometry analysis EWAT from IAV-infected mice (values for each individual mouse (#1 to 7), and means).

Besides, as stated in the Introduction, the differences between SCAT and EWAT have been repeatedly described (Reference 22), these fat depots notably differed regarding the adipocyte size: the adipocytes within SCAT are smaller than adipocytes within EWAT (Reference 28).

Thus, the comparison of adipocyte sizes between SCAT vs. EWAT from mock-treated and IAV-infected animals confirmed what is known from literature, that is the reason why these results are presented as Supplementary Fig. 2c.

The appearance of brown-like/beige adipocytes in fat tissues was only observed in SCAT and not in EWAT. This was confirmed by the increased number of small adipocytes in the SCAT of IAV-infected mice vs. the SCAT of mock-treated mice (Fig. 2b), the increased expression of the thermogenic, brown-specific, protein UCP1 in the SCAT of IAV-infected mice vs. mock-treated mice (Fig. 2c, 2d), and the increase local surface temperature (Fig. 2e). All these observations were absent when comparing the EWAT from IAV-infected mice vs. the EWAT from mock-treated mice.

We apologize if this was unclear. We therefore reformulate the text (Results), as follows:

Influenza infection is associated with subcutaneous adipose tissue browning. To determine whether SCAT and EWAT undergo qualitative and/or quantitative changes following influenza infection, we performed histologic and histomorphometric analyses at 7 dpi. The EWAT samples from mock-treated and IAV-infected mice had a similar histological appearance i.e. white adipocytes containing large unilocular lipid droplets (a typical WAT histology) (Supplementary Fig. 2a). In contrast, the SCAT samples from mock-treated and IAV-infected mice showed histological differences: numerous pockets of dense, small and multilocular adipocytes - a feature of brown-like adipocytes, also known as beige or brite adipocytes²⁷ - were found interspersed within the SCAT from infected animals (Fig. 2a). As reported in literature²⁸, histomorphometric analyses confirmed that adipocytes are smaller in SCAT than EWAT (Supplementary Fig. 2c). In accordance with the appearance of small brown-like/beige adipocytes, the SCAT from IAV-infected mice showed higher numbers of small adipocytes and reduced numbers of large adipocytes than the SCAT from mock-treated mice (Fig. 2b). The differences in adipocyte size between mock-treated and IAV-infected mice were less pronounced for the EWAT (Supplementary Fig. 2b). Moreover, levels of the mitochondrial uncoupling protein 1 (UCP1, a marker of the brown adipose tissue²⁹) were higher in SCAT protein lysates from infected mice than in those from mock-treated controls. In contrast, no UCP1 was detected in the lungs or EWAT lysates from IAV-infected mice (Fig. 2c, d). In line with increased levels of thermogenic UCP1, the surface temperature generated from the SCAT was higher in IAV-infected mice than in mock-treated mice (Fig. 2e).

These results emphasized the depot-specific nature of the effects of influenza infection on fat, with the appearance of thermogenic brown-like/beige adipocytes in the subcutaneous depot being the most notable.

Corresponding to the following revised Figure 2 and Supplementary Figure 2:

- Fig. 2c, 2d. (UCP1 expression, representative gel (2c) and quantification all gels (2d)).

In general, all bar graphs have either been converted to box-and-whisker plots, dot plots (with mean +/- SD (instead of SEM)), or dot-plots overlaid on the bars (+/- SD (instead of SEM)).

This change also concerned the data illustrating the increased expression of the thermogenic protein UCP1 in the SCAT upon influenza infection (combining the data from 4 Western-blot of protein extracts from 5 SCAT from mock-treated mice, 35 SCAT from IAV-infected mice, 5 EWAT from IAV-infected mice, and 5 Lungs from IAV-infected mice) showed on Fig. 2c. (representative blot), 2d. (quantification of all blots).

In addition, as requested from the Editorial Board, Western-blot 's crude data (gels and quantification) have been provided as Supplementary data.

Claim 3- IAV infection (of the host) leads to profound changes in the major WAT depots (i.e. the inguinal SCAT and the visceral (epididymal) adipose tissue (EWAT)) (Fig. 1)

In general, the statistics used have been detailed in the revised version of the manuscript (Materials & Methods section).

- Fig. 1b. (fat masses).

As requested, SCAT and EWAT masses from mock-treated vs. IAV-infected mice, formerly presented as bars, have been converted to box-and-whisker plots.

- Fig. 1c. (lipolysis-related gene expression).

The Reviewer referred to a publication (PMID: 21454566) reporting that *in vivo* deficiency of monoglyceride lipase decreased lipolysis and attenuated high-fat diet-induced insulin resistance. In this paper, no mention was made concerning WAT browning.

However, in the revised version of the manuscript, one reference on the role of lipolysis in proning SCAT browning has been added (Reference 68).

- Fig. 1d. (whole adipose tissue secretion analysis by the mean of tissue explant culture- remark on explant methodology).

Explant cultures of white adipose tissue are commonly used to analyze the secretory function of the adipose tissue since it preserves the physiological *in vivo* cross-talk between the numerous and various types of cells.

The reference of this method has been added (Reference 25), and the method has been more detailed in the Materials & Methods section.

Besides, lymph nodes associated to the fat tissues were excluded at the time of harvesting, so that the immune cells that are present in the adipose tissue explant were either resident immune cells or infiltrated immune cells.

- Fig. 1d. (whole adipose tissue secretion analysis by the mean of tissue explant culture- remark on leptin secretion).

As stated in the Introduction, the differences between SCAT and EWAT have been repeatedly described, these fat depots notably differed regarding their secretory functions (e.g. Reference 22).

Here, we intended not to compare SCAT and EWAT in general but to compare the SCAT's response to IAV infection vs. the EWAT's response to IAV infection. Compared to mock-treated controls, infection was associated to increased secretion of leptin by the SCAT, and to decreased secretion of leptin by the EWAT (while SCAT and EWAT both enhanced their production of IL-1 β , IL-6 and IL-10 upon infection).

We apologize for this misunderstanding, and we reformulated this part of the Results section as follows:

*Adipose tissue is composed of several cell types that include mature adipocytes, preadipocytes, and a range of innate and adaptive immune cells¹⁷⁻²⁰. As an endocrine organ, WAT secretes numerous cytokines, metabolites and hormones that originate from all these cell types¹⁸. By preserving the physiological *in vivo* cross-talk between cells, explant cultures of WAT allow analyzing the secretory function of the whole tissue²⁵. We therefore used tissue explant cultures to investigate whether influenza infection was associated with changes in SCAT and EWAT's*

secretory function (Fig. 1d). When compared with controls, the adipose tissues collected from infected mice produced higher levels of the proinflammatory cytokines IL-1 α and IL-6, and of the anti-inflammatory cytokine IL-10. Following IAV infection, the release of leptin - a WAT-derived hormone that regulates feeding behavior, body weight²⁶, and innate and adaptive immune responses²¹ - was enhanced for SCAT and decreased for EWAT.

Claim 4- IAV infection rewires energy metabolism (transcriptomics):

As stated by the Reviewer, transcriptomic analyses on whole fat tissues correspond to a mixture of numerous, and different cell-types that composed the tissue. As suggested, the revised version of the manuscript now emphasized more the heterogeneity of the adipose tissue, regarding its cellular composition.

The text has been changed according to this essential comment, as follows:

- Results-Fig. 1.

Adipose tissue is composed of several cell types that include mature adipocytes, preadipocytes, and a range of innate and adaptive immune cells¹⁷⁻²⁰. As an endocrine organ, WAT secretes numerous cytokines, metabolites and hormones that originate from all these cell types¹⁸. By preserving the physiological in vivo cross-talk between cells, explant cultures of WAT allow analyzing the secretory function of the whole tissue²⁵. We therefore used tissue explant cultures to investigate whether influenza infection was associated with changes in SCAT and EWAT's secretory function (Fig. 1d).

- Results-Fig. 5.

WAT's cellularity is highly heterogenous^{18-20,22}. Mature, lipid-filled adipocytes make up approximately 30% of adipose tissue cells. The remaining stromal vascular fraction (SVF) cells include e.g. vascular cells, fibroblasts, innate and adaptive immune cells, and preadipocytes. We therefore separated mature adipocytes from SVF cells through enzymatic fractionation of the SCAT and EWAT from mock-treated and IAV-infected mice. As shown in Supplementary Fig. 5a, viral RNA was detected in SVF cells isolated from SCAT and not in SVF cells isolated from EWAT. No viral RNA was detected in adipocytes isolated from either fat depot (not shown).

- Discussion.

In rodents, SCAT (less frequently EWAT) is readily able to convert to a brown-like state in response to environmental stimuli including chronic cold adaptation, exercise and nutritional challenges; as well as external and internal cues such as pharmacological treatment with α -adrenergic receptor agonists and various peptides and hormones⁷¹. Leptin (which secretion is enhanced in the SCAT upon infection) and insulin (which blood level is increased in infected mice) have been reported to promote WAT browning⁷². Influenza infection may also have favored the accumulation of immune cells associated with WAT browning, such as M2 macrophages, eosinophils or innate lymphoid type 2 cells⁷³, in the SCAT.

Strikingly, we found that immune cells harboring viral proteins (HA) were present in the WAT (mostly SCAT) of influenza-infected mice. Immune cell egress from the lungs has been reported in the context of influenza infection^{74,75}. Thus, immune cells (either infected or harboring remnant viral antigens and RNA) may have entered the bloodstream through the capillaries of damaged alveolar wall of the infected lungs and reached fat depots, participating to the observed changes in WAT's secretory function and to the induction of antiviral-innate-immunity-related pathways. The phenotypic characterization of immune cells that are present in the fat depots of IAV-infected mice is currently under investigation.

Alongside immune cells, viral antigen-positive preadipocytes were also found, although less frequently, in the WAT from infected mice. How preadipocytes acquired viral antigens from immune cells that have migrated from infected lungs to WAT, remains to be established. Also, determining whether other WAT's cell types are impacted during influenza infection should be investigated since WAT's functionality result from the concerted interplay between all

these cells. Single-cell RNA sequencing combined with flow cytometry or mass cytometry (CyTOF) could help answer this key question⁷⁶.

Besides,

1. We agree that our work did not address the effect of influenza infection on fat tissue, in a cell-intrinsic way. To study cell intrinsic changes in adipose tissue, one should isolate adipocytes and preadipocytes (but also the different innate and adaptive immune cells) from the fat tissue of IAV-infected mice and do bioenergetics on these cells. As stated in the revised Discussion, technologies such as single-cell analyses would help answer this key question,

2. Transcriptome analyses did not intend to compare the transcriptomes from SCAT vs. EWAT (it is known that they differed (Reference 22)) but to compare the transcriptome from SCAT mock vs. SCAT IAV; and the transcriptome from EWAT mock vs. EWAT IAV. We apologize if this was unclear. This part has also been re-written and shortened, as follows:

To gain insights into the molecular basis of the depot-specific effects of influenza infection on fat tissues, we performed transcriptomic analyses on the SCAT and EWAT isolated from mock-treated and IAV-infected mice ($p \leq 0.05$ and fold change (FC) cutoff of ≥ 2). In SCAT, 1,214 genes were found differentially expressed between mock-treated and IAV-infected mice. In EWAT, 660 differentially expressed genes (DEGs) were identified. Venn diagrams were used to display comparison of the lists of upregulated or downregulated DEGs (Fig. 3a). Upon infection, 296 genes were upregulated in both SCAT and EWAT, 268 genes were upregulated only in SCAT, and 177 genes were upregulated only in EWAT.

3. Nota bene: We agree that the throughout analysis of the transcriptomic data should have been done at the first place. Yet, we first used free programs (GO Resources) to analyze our transcriptomic data, we only had access to Ingenuity Pathway Analysis (IPA) later.

Claim 5- “novel aspects of the metabolic control of influenza virus-host interactions at the molecular, cellular and tissue levels.”- (Discussion)

We agree that speaking of IAV/host interactions at the organism, tissue, and cell level was overreaching. All changes made in the revised version of the manuscript took this specific comment into account.

6. Minor claims

i. “influenza infection has a long-lasting impact on whole-body glucose metabolism, and that the impact is independent of the host’s dietary status.”- Fig. 4c.

We thank the Reviewer for having pointed out this omission. The group of IAV-infected standard-diet-fed mice has thus been added in revised Fig. 4, as follows:

Fig. 4 Influenza infection alters the host’s energy metabolism in a lasting manner. **a** Blood levels of glucose (10-12 hours fasting), insulin (10-12 hours fasting) and resistin (fed state) in mock-treated and IAV-infected mice at 7 dpi, $n = 15-20$ mock and $n = 15-30$ IAV. Individuals values, as well as means \pm SD are shown. $p < 0.01$, **** $p < 0.0001$. **b** Blood levels of glucose (10-12 hours fasting), insulin (10-12 hours fasting), and resistin (fed state) in mock-treated and IAV-infected mice at week 20 post-infection, $n = 8$ mock and $n = 6$ IAV. Individuals values, as well as means \pm SD are shown. $p < 0.05$, * $p < 0.01$. **c** Body weight change (% initial body weight) of mock-treated and IAV-infected mice fed with standard diet (SD) or with high-fat diet (HFD) from 7 dpi (arrow) (respectively: mock SD, IAV SD, mock HFD and IAV HFD), $n = 5$ mock SD, $n = 10$ IAV SD, $n = 10$ mock HFD and $n = 15$ IAV HFD. Data are expressed as mean \pm SD. The corresponding area under the curve (AUC) is shown in Supplementary Fig. 4b. **d** Intraperitoneal glucose tolerance test (IP-GTT) was performed in 10-12 hours-fasted mice, 18 weeks post-infection. Blood samples for glucose level determination were taken from the tail vein before glucose administration (i.p. injection, 1 g/kg BW) and after 15, 30, 60, 120 and 180 minutes, $n = 5$ mock SD, $n = 10$ IAV SD, $n = 9$ mock HFD and $n = 12$ IAV HFD. Data are expressed as mean \pm SD. The corresponding area under the curve (AUC) is shown in Supplementary Fig. 4c. **e** Blood levels of glucose, insulin and resistin (10-12 hours fasting) in mice, at sacrifice, $n = 6-7$ mock HFD, $n = 11-12$ IAV HFD and $n = 3-4$ mock SD. The dotted lines indicated the values obtained for mock-treated standard diet-fed animals. Individuals values, as well as means \pm SD are shown. $p < 0.05$, ns = not significant.

ii. IAV changes host’s glucose metabolism that persist long after infection has been cleared and IAV alters whole- body glucose metabolism long after resolution (>2wk). The association between blood glucose levels and IAV infection have been reported, indeed in diabetic patients (with high blood glucose levels) the severity of infection is more pronounced than in normoglycemic non-diabetic individuals. Interestingly enough, a publication reported that insulin treatment of diabetic mice can ameliorate Flu. So, the fact that diabetes and obesity predispose to increased flu severity is well-known. However, the other way, *i.e.* the impact of influenza infection on the host’s glucose metabolism has scarcely been investigated.

Are the claims novel? In order to be more accurate in the already known impact of viral infections on the adipose tissue, we modified the text accordingly and added several references.

REVIEWERS' COMMENTS:

Reviewer #1 (Remarks to the Author):

The authors addressed the main concerns from the reviews, the revised version of the manuscript appears to be good except a minor revision:

-The PCR reaction conditions should be reported with cycles.

COMMSBIO-19-1556A_AYARI et al.**Responses to Reviewer's last minor request**

First, we thank the reviewer for the constructive comments and suggestions. The manuscript has significantly gained in precision and clarity.

The remaining request was:

*The authors addressed the main concerns from the reviews, the revised version of the manuscript appears to be good except a minor revision:
-The PCR reaction conditions should be reported with cycles.*

To fulfill this request, Figure 5a, Figure 5b, Supplementary Figure 5a, Figure 6a, and Supplementary Figure 7b (all of which presenting data of the viral load in tissues or cells, evaluated by PCR) have been changed. Results are now expressed as Cycle Threshold (Ct) values. The text in the Results section as well as legends to Figures, legends to Supplementary Figures, and the text in the Methods section have been changed accordingly.